# Spike conformational and glycan heterogeneity associated with furin cleavage causes incomplete neutralization of SARS-CoV-2

Sahil Kumar[1,11,12], Rathina Delipan[1,12], Chanchal Sharma[1], Jyoti Jadoun[1], Kawkab Kanjo[2], Randhir Singh[3], Raju Rajmani[2], Suprit Deshpande[4], Rajesh Pandey[5], Krishan G. Thakur [1,6], Jayanta Bhattacharya[4], Rogier W. Sanders [7,8,9], Marit J. van Gils [8,9], Raghavan Varadarajan [2,10] & Rajesh P. Ringe[6] ✉

SARS-CoV-2 Spike - the sole neutralization target, is highly resilient to the immune pressure driving genetic evolution. While potency and breadth of neutralization are widely studied, the incomplete neutralization - the mechanism of resistance without needing genetic change - remains unexplored. Several monoclonal antibodies, although potent, showed incomplete neutralization of genetically homogeneous pseudovirus suggesting the existence of distinct spike conformations. The residual infectivity at high antibody concentration indicates a viral fraction with intrinsic resistance to the antibody. Although the published studies on spike glycosylation, structure, and conformations provide evidence of spike heterogeneity the precise mechanism for the incomplete neutralization has not been established. In this study, we devise a method to separate the un-neutralized virion population, called as persistent fraction of infectivity (PF), and characterize the viral spike protein. The neutralization resistance of PF is stable and unrelated to the conformational equilibrium that exists in the pseudovirus stock. The spike on the PF is highly cleaved between S1 and S2, adopts the closed conformation, and express more mannosidic glycans on RBD than the total virus population. Our study provides possible explanations for the incomplete neutralization by antibodies and delineates the association between furin cleavage of spike, its conformation and glycosylation.

Viral neutralization is a key component of immunity, and antibodies that specifically target the SARS-CoV-2 spike protein and block viral entry are particularly significant. Their effectiveness is dependent on three key factors, i.e., potency, breadth, and completeness of neutralization. While potency and breadth are highly valued, completeness of neutralization—or the neutralization efficacy—is rarely taken into account. In the standard neutralization assay using pseudovirus or the authentic virus, many nAbs, including a clinically used S309 (Sotrovimab), show saturation of neutralization below 100%[1–3]. At high antibody concentrations, the unneutralized fraction of

virions, called the persistent fraction (PF), is responsible for residual infectivity. In principle, higher PF would reduce the therapeutic efficacy of nAbs and vaccine-induced nAbs. In the presence of potent nAbs, the PF acts as a source of continued infection, can alter the viral dynamics in vivo, enabling sustained viremia that serves as a reservoir for genetic variations[4,5]. Indeed, S309-treatment in humans reduced the viremia, but resistant viruses rapidly emerged[3,6]. Similarly, multiple studies have reported the breakthrough infections in the vaccinated or recovered individuals who had neutralizing antibodies against the infecting strain[7–10]. These observations give credibility to the theory that a fraction of viruses in the aerosols escape neutralization by host antibodies and establish a breakthrough infection.

The recent studies on HIV have demonstrated that PF is caused by glycan and conformational heterogeneity of gp160[11,12]. Glycosylation, mediated by multiple enzymes, is influenced by the steric accessibility of the glycosylation sequons. The sequons with limited steric access are often under-processed and tend to retain high-mannose type glycans, whereas those that are more accessible are more likely to be processed into complex-type glycans. Although PF has not been studied for SARS-CoV-2, data from multiple studies indicate that it is present in genetically homogeneous viruses[13–15]. This implies that post-translational modification (PTM) of the Spike can modulate its conformation and antigenicity. Lu et al. demonstrated four major conformations based on single-molecule Förster resonance energy transfer (smFRET) analysis of spike on the pseudovirus particles, and most closed conformations were associated with low neutralization sensitivity[16]. Thus, structural heterogeneity in viral glycoproteins is the first line of defense against nAbs, which enables virus infection until genetic modifications emerge to counter the sustained neutralization pressure[17,18]. The ability to survive under neutralization pressure, therefore, has implications in recurrent and persistent infections[19].

Although the high-resolution cryo-EM structures of stabilized ectodomain or on-virus Spike provide insights into structural organization and its evolution with genetic modifications, they do not provide as much information regarding the microheterogeneity at the epitope level influenced by Spike's glycosylation and other post-translational modifications (PTMs)[20–27]. The real-time structural dynamics of the on-virus spike also confirmed distinct conformational states matching cryo-EM data[16.] However, neither the underlying regulatory mechanism of intrinsic antigenic heterogeneity nor its relationship with incomplete neutralization is clear. Similarly, the studies that characterized glycans on viral Spike did not provide insights as to how glycan heterogeneity modulates viral characteristics[28,29], although a recent study shed some light on the modulation of neutralization efficacy by monoclonal antibodies after deleting a glycan from Asn343 in RBD[30]. The purpose of this work, therefore, was to examine the incomplete neutralization and the characteristics of PF responsible for residual infection.

Here, we show that SARS-CoV-2 evaded antibody-mediated neutralization using conformational and glycan heterogeneity of the Spike protein. Being a crucial mechanism of immune evasion prior to the genetic adaptations, incomplete neutralization has implications on the development of therapeutic antibodies and vaccine design. Here, we developed an affinity chromatography-based method to enrich PF from the total virus population and characterized on-virus Spike glycosylation and conformations to explain intrinsic neutralization resistance.

## Results

### The estimation of persistent fraction of SARS-CoV-2 in neutralization

The persistent fraction of infectivity (PF) is the residual infection caused by non-neutralizable virions in the presence of a high concentration of antibodies. We first examined whether PF arises due to depletion of antibody by formation of virus-antibody complexes,

leaving a fraction of virions unbound when a high pseudovirus inoculum is used. We chose three nAbs, S309, COVA1-16, COVA2-17 to assess this against serial dilutions of pseudovirus inoculums. At a fixed excess concentration of each neutralizer (50 μg/ml), the percent neutralization of WT hinged around 80%, ~98%, and 75% respectively (i.e., 20%, 2%, and 25% PF respectively), and that remained constant with changing virus inoculum over a 100-fold range (Supplementary Fig. S1A). The percent residual infection (PF) remained well above 1%, a threshold considered in this analysis, for S309 and COVA2-17 on the log y-axis. However, when the PF was much smaller, such as in the case of COVA1-16, it was more consistently recorded at a higher virus titer and could be differentiated from the standard error of mean (Supplementary Fig. S1B). This data showed that smaller persistent fractions could be less easy to measure accurately when low virus titers are used. Therefore, for all subsequent assays, we used at least $3 \times 10^5$ RLU of pseudovirus to rule out the standard error and confidently estimate the size of the PF. This result confirmed that the PF does not arise due to excess antigenic load in pseudovirus neutralization but adheres to the percentage law of neutralization[31]. Next, we assessed the effect of major viral entry receptors on the viral neutralization and the PF. The receptor stoichiometry can influence the viral entry and neutralization. The major receptor ACE2, and minor receptor Neuropilin-1 (NRP1), and protease TMPRSS2, were considered for this analysis. The HEK293 cells expressing ACE2, ACE2/TMPRSS2, or ACE2/TMPRSS2/NRP1 were used to measure the neutralization of WT by RBD-specific nAbs (COVA2-15, S309), and NTD-specific nAb COVA2-17. COVA2-15 showed complete neutralization in all three cells, but potency decreased in cells that co-expressed TMPRSS2 and NRP1. Similar trends of decreasing potency were also seen for COVA2-17 and S309, but there was a noticeable increase in PF when NRP1 and TMPRSS2 was co-expressed on the cells, suggesting that the receptor expression profile on the target cells can influence the PF (Fig. 1A). The same experiment was also conducted using authentic SARS-CoV-2 virus which showed broadly similar pattern (Fig. 1B). We also examined whether the PF appears in other cells. The neutralization of WT PV was examined in Vero-E6 and A549 cells, which naturally support SARS-CoV-2 infection. In both cell types, the S309 and COVA2-17 showed incomplete neutralization, while COVA2-15 fully neutralized the virus (Supplementary Fig. S1D). These data confirmed that PF is related to antigenic variation of Spike and is manifested by both the pseudotyped virus and the authentic virus. Since ACE2 and TMPRSS2 are the most important entry factors and cells expressing them are widely used for neutralization assays, we used 293T-ACE2/TMPRSS2 cells for all subsequent experiments pertaining to PF.

To measure the PF in neutralization, we used nAbs specific to dominant neutralization targets - RBD and NTD, and the major SARS-CoV-2 variants – WT, Alpha, Beta, Delta, and BA.5. The respective pseudoviruses were variably neutralized by the nAbs, showing differences in potency and the size of PF. Overall, WT was more sensitive and showed smaller PF than subsequent variants (Fig. 1C, D). For instance, S309 neutralized WT, Alpha, Beta, and Delta within a short-range difference in potency (IC50 0.02-0.07 μg/ml), but the efficacy of neutralization was highest for WT than others (Fig. 1C, D). The potency of ACE2-Fc differed among the variants, but the overall efficacy of neutralization remained similar at ~100% (Fig. 1C). The PF was also confirmed by residual infection of GFP-expressing pseudovirus in the presence of excess concentration of antibody (Supplementary Fig. S1 C).

We also analyzed the PF in the neutralization by human convalescent sera from the first wave of the COVID-19 pandemic caused by the Wuhan strain of SARS-CoV-2 in northern India. The first wave spanned the period from May 2020 to February 2021, and the sera were obtained from the individuals after 2-4 weeks post-recovery from COVID-19. Several sera had weak neutralization activity, failing to produce a complete neutralization curve and were excluded from the

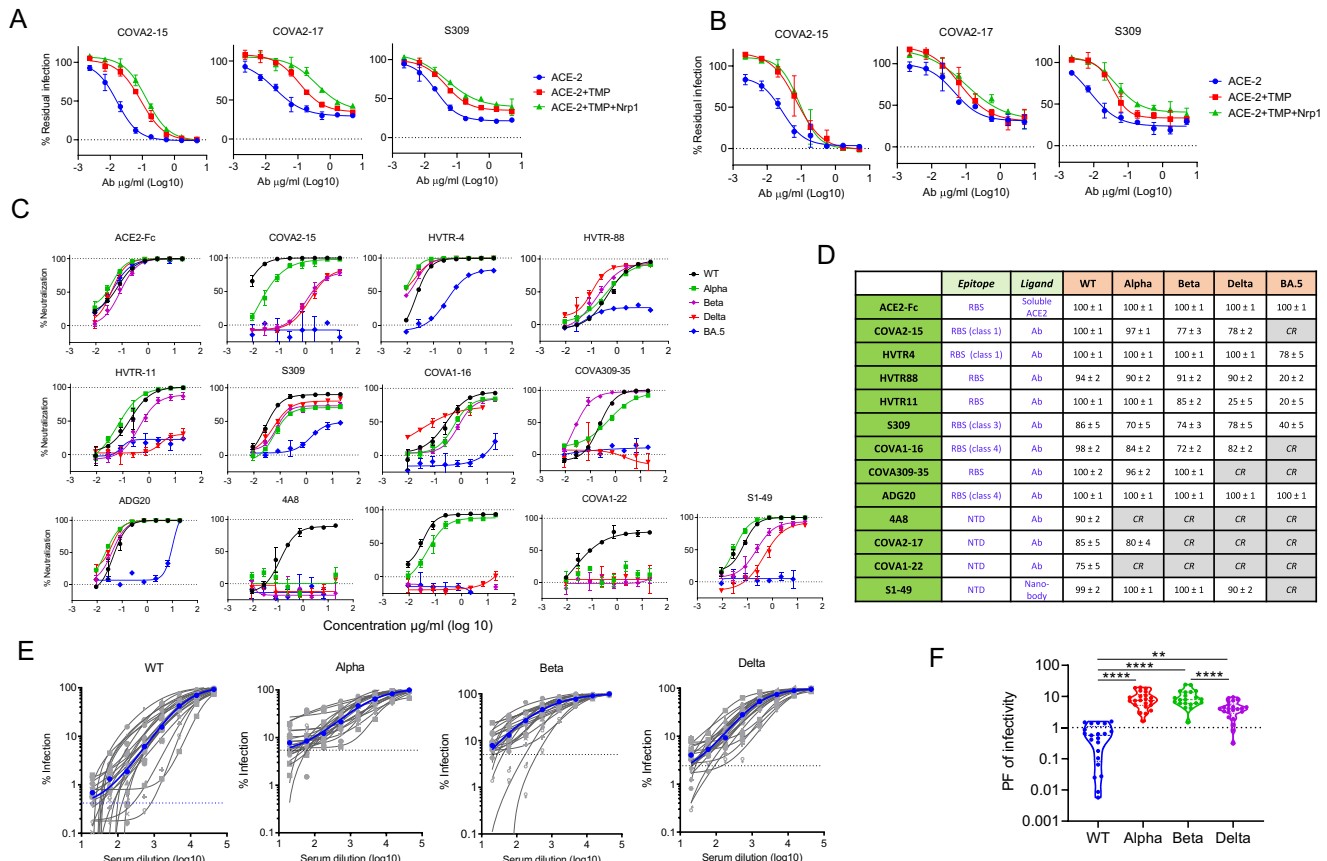

**Fig. 1 | Neutralization pattern and establishing persistent fraction of SARS-CoV-2 strains in the neutralization assay. A** The neutralization pattern of COVA2-15, S309, and COVA2-17 against WT pseudovirus in HEK293T cells expressing ACE2 (blue), ACE2/TMPRSS2 (red), or ACE2/TMP/NRP1 (green) receptors. Data are presented as mean values +/− SD (**B**). The neutralization pattern of the authentic virus (WT) by COVA2-15, S309, and COVA2-17 in the same cells as in (**A**). **C** The neutralization pattern of various pseudoviruses against indicated antibodies, hACE2-Fc, and nanobody S1-49. The assays in A-C were done in two technical repeats. Data in A-C are presented as mean values +/− SD (**D**). The maximum neutralization values calculated from data in B for the indicated pseudoviruses and the standard deviation in maximum neutralization is based on three independent experiments. The epitope classes for receptor binding site (RBS) antibodies in RBD are indicated by sub-sites for those whose precise site is defined. CR is an acronym for complete

resistance and indicated if virus is completely resistant to the given antibody. In such cases the persistent fraction is 100%. **E**. The percent residual infection of pseudovirus in the neutralization assay is plotted as a function of serum dilution. The blue line represents the median neutralization curve derived from all the sera (*n* = 22) tested, and the dotted line below the median neutralization indicates the saturation point of maximum reduction of infection. **F** The PF of infection of indicated pseudoviruses against the sera in E (*n* = 22) is shown as percent infection in the presence of the highest concentration of serum antibodies. The statistical difference was calculated using the Kruskal-Wallis test, and the multiple comparison between the pseudoviruses was based on Dunn's post-test multiple comparison. Adjusted *P*-values < 0.0001 (****), < 0.0019 (**), and 0.02 (*) are indicated. All data is based on at least two independent repeats. All statistical analyzes were done on measurements done on distinct samples.

---

analysis of PF. Only 20/121 sera were potent, reaching the maximum neutralization plateau of WT, Alpha, Beta, and Delta (Fig. 1E). For WT, the median residual infection at lower serum dilutions plateaued below 1%, whereas for Alpha, Beta, and Delta, it did at approximately 4%, 9%, and 3% respectively (Fig. 1E, F). Thus, the mutations in natural variants significantly increased the PF and decreased the potency of monoclonal and polyclonal serum antibodies (Fig. 1E, F). This, however, is a rough estimation as a few out of 20 sera did not reach a stable neutralization plateau against VOCs, which may affect the precise estimation of PF. Nonetheless, the trend was clear from the median curve and the statistical differences that emerged from this less precise PF estimations.

## Isolation of PF by depleting antibody-sensitive viruses from original pseudovirus population

The detailed investigation on the PF, including virological and biochemical characterizations, would be possible if the PF is separated from the total population of virus particles. We employed a virus depletion technique by which Ab-sensitive virions could be largely eliminated to enrich PF as described earlier[11,12]. Two methods were

tested to deplete the pseudovirus by using S309 as a depleting antibody, which neutralized WT by ~80% (i.e., ~20% PF). In the first, we prepared the affinity column containing S309-coupled sepharose beads making a thick bed (Fig. 2A). The transfection supernatant containing pseudovirus was incubated with beads on a nutator for 2 h at room temperature to adsorb the virions (solid phase depletion), followed by flowing the supernatant slowly through the layer of settled beads. The flowthrough containing the non-captured virions (i.e., PF) was collected and stored at − 80 $^0$C for further use (Fig. 2A). To generate an undepleted virus control, the same procedure was followed but with antibody-free beads. The second protocol was based on depletion in the solution phase, in which the pseudovirus stock was first incubated with the antibody. We used a 10 to 20-fold excess antibody of the minimum concentration required to achieve maximum neutralization. For example, S309 attained maximum neutralization at 0.5 µg/ml in the standard neutralization assay we used 10 µg/ml for virus depletion. As we used undiluted virus supernatant freshly harvested from the transfected cells this antibody concentration was still sub-saturating as the virus still showed ~10% residual infectivity (Supplementary Fig. S1C). After incubating S309 with the virus for

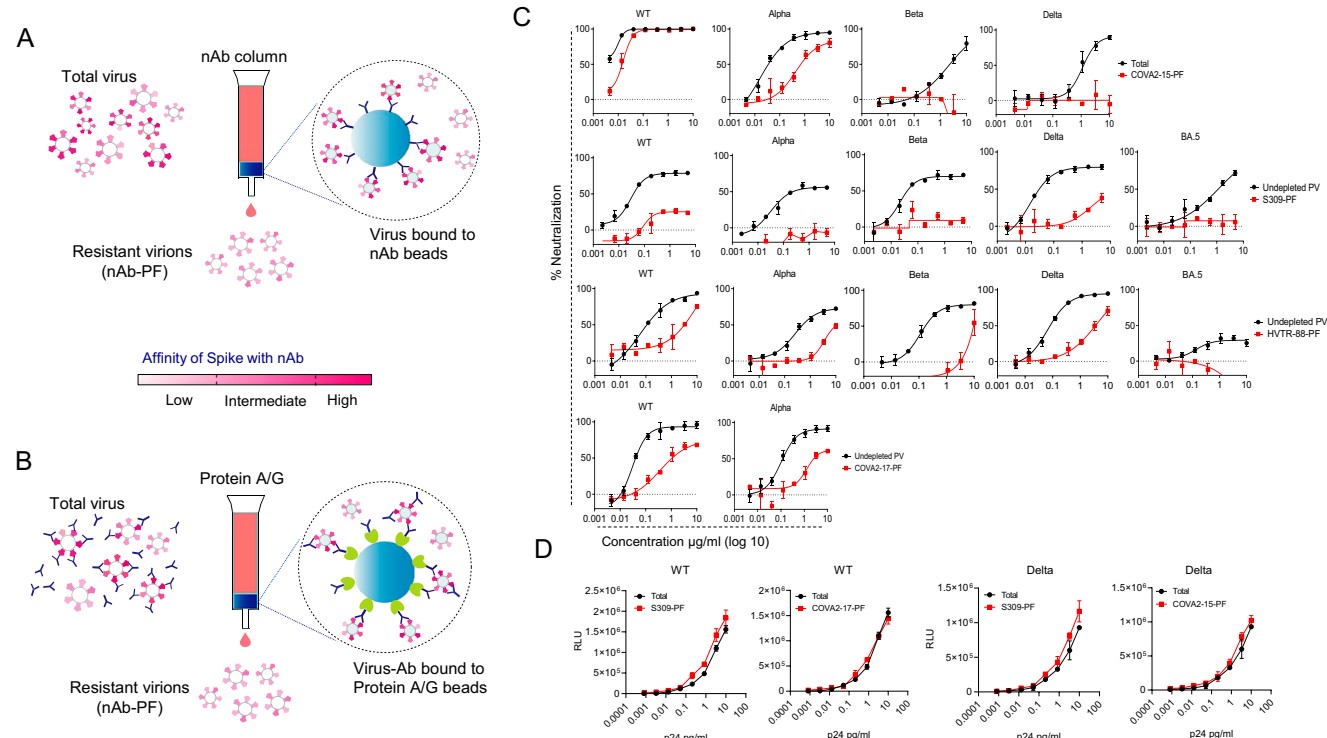

**Fig. 2 | The depletion of Ab-sensitive viruses from the total population to isolate PF. A** The method of pseudovirus depletion using Ab-coupled beads. The total virus population was incubated with Ab-beads in the column to capture Ab-sensitive virions. The flowthrough containing Ab-resistant virions was collected as PF. **B** Virus depletion by incubating transfection supernatant with antibody, followed by capture of virus:Ab complexes on the Protein A/G bed to obtain PF in the flowthrough. The color key depicts the affinity differences between viral spikes for the capture Ab that determines the virus depletion. **C** Neutralization profile of total pseudovirus population (Total) and Ab-depleted population (PF). The total virus is mock-depleted virus, which was obtained by the same depletion procedure in (**B**) but without using any depleting Ab. The PF of pseudovirus was labeled after depleting the antibody, e.g., PF obtained by S309 was labeled as S309-PF. **D** Infectivity of total pseudovirus population and PF. The p24 antigen quantification of the pseudovirus stocks was performed using serially diluted viral supernatants. The same serially diluted virus supernatants were also used to infect 293T-ACE2/TMPRSS2 cells, and the infection was recorded as relative luminescence unit (RLU). The data is representative of two independent repeats.

30 minutes, the virus:Ab mixture was passed through a protein-A/G column to remove free antibody and antibody-virus complexes via Fc (Fig. 2B). This step was repeated with fresh protein-A beads to remove any remaining antibodies or complexes. To obtain a negative control, the same procedure was performed without adding S309. The S309-PF (S309-depleted pseudovirus) isolated from both the methods showed a substantial decrease in sensitivity to S309 but solution-phase depletion was more effective in enriching the PF (Supplementary Fig. S2A). We assessed whether PF obtained from solution phase depletion contains any residual antibody used for depletion. In ELISA, the presence of virions was confirmed by detecting the viral spike, but depleting antibody - S309, was un-detectable (Supplementary Fig. S2A). This data confirmed the depleted virus is pure and free of antibodies. Considering the dynamic conformational equilibrium of viral spike in the virus supernatant, the neutralization of PF was assessed after incubating the depleted virus at 37 °C for up to 6 hours or stored at −80 °C for up to 12 months before performing neutralization assay. The neutralization resistance of PF did not decrease by these treatments, suggesting that the PF represents an intrinsically resistant fraction of viruses and is not related to dynamic conformational fluctuations of viral spike (Fig. S2C, D). Based on these findings, we chose solution-phase depletion to obtain PF using an authentic virus. We used WT and Delta virus and depleted them by COVA2-15, S309, or HVTR88. The depleted virus was substantially more resistant to the depleting antibodies, corroborating the results obtained using pseudoviruses (Supplementary Fig. S3A, B). The depletion method employed a sub-saturating concentration of Ab, and the binding

kinetics favor neutralization of more sensitive virions and subsequent capture on the affinity column. Conversely, virions with low-affinity spike variants are less likely to bind and therefore pass into the flowthrough. Although still sensitive to depleting antibody, the sensitivity of PF was substantially reduced, making the method suitable to enrich the PF, though it cannot be used to exclusively isolate pure PF. The data revealed that both the virus types exhibit the antigenic heterogeneity of Spike quite similarly. Therefore, for all subsequent analyses we used pseudoviruses as it fully ensures genetic homogeneity of Spike.

**Existent in the initial population, the PF is inherently resistant to the depleting nAb and represents distinct Spike conformations**
To gauge the antigenic heterogeneity of epitopes in NTD and RBD, we chose representative nAbs for depletion of nAb-sensitive virions in order to enrich the PF. Accordingly, we selected COVA2-15, S309, HVTR88, and COVA2-17, which showed various PFs against SARS-CoV-2 strains. The depletion by each antibody removed most of the sensitive virions from the original populatio,n leaving the PF with significantly reduced infectivity. The loss of infectivity was dependent on the antibody's efficacy of neutralization against a particular variant. The depletion by COVA2-15, which is highly potent against WT and Alpha with very less PF in neutralization, led to the loss of infectivity by ~ 90% but S309 showing higher PF reduced the infectivity by an average of ~ 70% (Supplementary Table S1). The loss of infectivity was due to a reduced number of virus particles. To check the intrinsic infectivity of PF, we assessed the infectivity of PF and the total pseudovirus

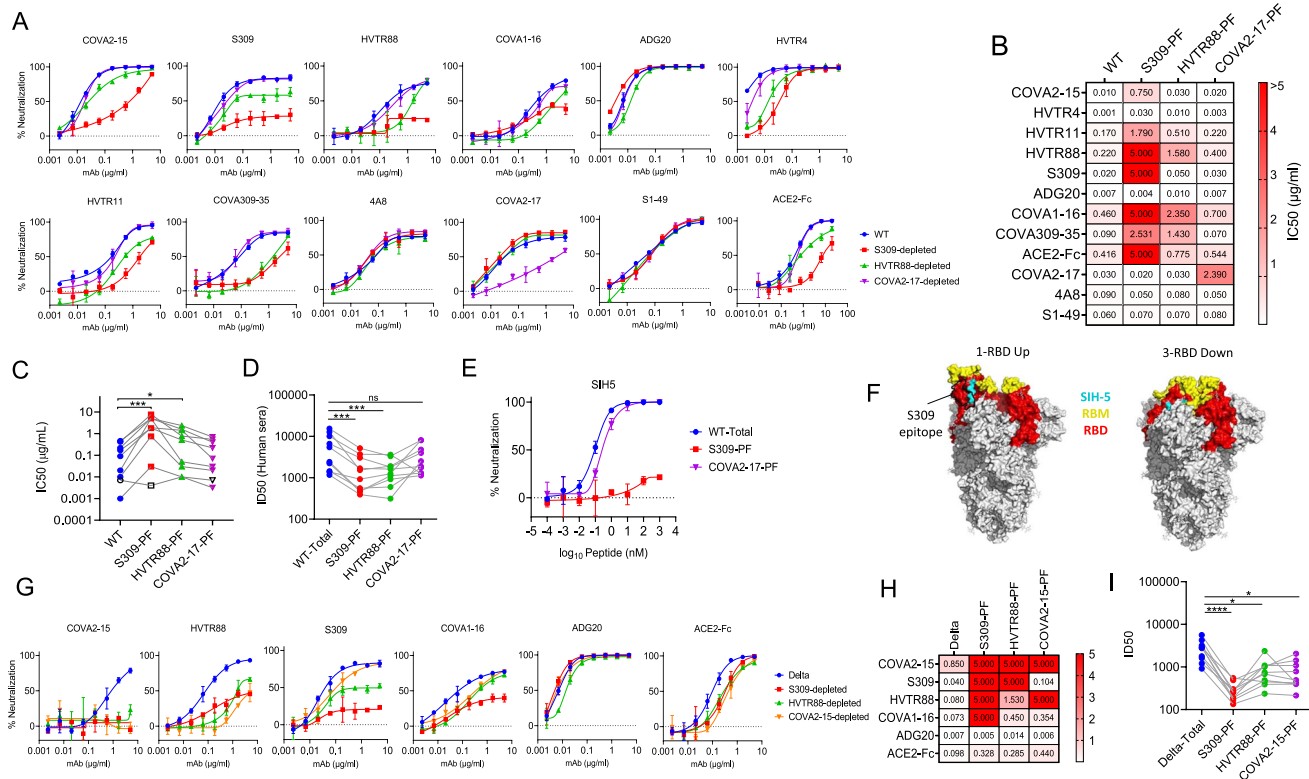

**Fig. 3 | Topological analysis of antigenic landscape of Spike expressed on the PF. A** Neutralization sensitivity of WT pseudovirus is plotted as a function of Ab concentration using an asymmetric sigmoidal function of non-linear regression. The neutralization curves of WT-total virus (blue), S309-PF (red), HVTR88-PF (green), and COVA2-17-PF (purple) against the indicated nAbs are shown. The data is representative of two biological repeats. Data are presented as mean values +/− SD (**B**). The IC50 values in μg/ml of WT pseudovirus derived from data in A. The color key is shown to depict the IC50 range, and the darkest shade of red shows IC50 ≥ 5 μg/ml. **C** Neutralization sensitivity of WT pseudoviruses in A to RBD-binding antibodies ($n = 8$) and ACE2-Fc. The differences in sensitivity were calculated using Friedman's test, and multiple comparison between undepleted and depleted pseudovirus was done by Dunn's multiple comparison test with adjusted $P$-values indicated as * ($P < 0.05$), ** ($P < 0.009$), *** ($P < 0.0009$), and **** ($P < 0.0001$). **D** Neutralization sensitivity of WT pseudoviruses in A to human convalescent sera from the first wave of COVID-19 ($n = 10$). Each serum sample was

obtained from a distinct individual. The median ID50 titer for total virus compared with PFs by using the same statistical methods and adjusted $P$-values mentioned as in 3 C. **E** Neutralization sensitivity of WT to SIH-5 peptide, which targets RBD in "up" conformation. The data is representative of two biological repeats. Data are presented as mean values +/− SD. **F** A structural model of Spike ectodomain with 1-RBD "up" (6VYB) and 3-RBD "down" (6VXX) conformations[62]. The RBD (red), receptor binding motif (yellow), and SIH-5 peptide binding residues (cyan) are shown. The S309 binding site is outlined in black. **G** Neutralization pattern of undepleted Delta pseudovirus (blue), S309-PF (red), HVTR88-PF (green), and COVA2-15-PF (orange) against various RBD-specific nAbs. The data is representative of two biological repeats. Data are presented as mean values +/− SD. **H** The IC50 values in μg/ml from the data in (**G**). **I** Neutralization sensitivity of Delta to the same sera as in D. All neutralization experiments were done in two technical repeats. The statistical analysis between the undepleted Delta pseudovirus and its PF was calculated using Dunn's multiple comparison test. The adjusted $p$-values are indicated as in 3 C.

---

population against the p24 antigen. The intrinsic infectivity of PF of WT and Delta was similar to the corresponding total virus population (Fig. 2D). Despite eliminating a large fraction of virions by depletion process, the infectivity of PF was sufficient to reliably assess the neutralization sensitivity in the standard assay.

S309 recognizes a peptidoglycan epitope including a glycan at N343 and its core epitope was conserved in all the variants tested here (Fig. 2C). The S309-PF of each viral variant was markedly resistant to S309 compared to undepleted virus (Fig. 2C). Other RBD-specific antibody COVA2-15 recognizing an epitope in receptor binding motif (RBM) was potent against WT and Alpha but less so against Beta and Delta due to T478K or E484K mutations. COVA2-15, showing complete neutralization of WT could not enrich PF very effectively and the PF was only slightly resistant to COVA2-15. However, the COVA2-15-PF of Alpha, Beta, and Delta were markedly resistant to COVA2-15, suggesting greater heterogeneity of COVA2-15 epitope in these variants than WT Spike (Fig. 2C). Likewise, the PF was obtained using HVTR88 - an RBD nAb which showed incomplete neutralization of all five variants. Their PFs showed marked resistance to the same antibody (Fig. 2C). An NTD-specific nAb COVA2-17 was active only against WT and Alpha, showing saturation of neutralization below 70% but was inactive

against other variants of concern (VOC), therefore, the PF was obtained only from WT and Alpha. The COVA2-17-PFs were substantially resistant to the same nAb suggesting the heterogeneous nature of its epitope in NTD (Fig. 2C). Overall, the PF was substantially resistant to the depleting nAb and the difference in sensitivity between undepleted virus and its PF was higher in case of Beta, Delta, BA.1, and BA.5 compared to WT and Alpha (Fig. 2C).

## The antigenic landscape of the RBD of PF is distinct from the total virus population, conferring broad resistance to neutralization

The isolation of PF facilitated the investigation of antigenic heterogeneity and conformational flexibility of the epitope clusters located within the NTD and RBD. We assessed the neutralization sensitivity of PFs to nAbs specific to RBD and NTD. As anticipated, S309-PF exhibited a loss of sensitivity to S309, resulting in a significant fall in maximum neutralization plateau to 25% from 67% (Fig. 3A). In addition, PF showed resistance to other RBD-specific neutralizers, including COVA2-15, HVTR11, HVTR88, COVA1-16, COVA309-35, HVTR4, and ACE2-Fc. The PF was somewhat equally sensitive to ADG20 (Fig. 3A, B). Likewise, HVTR88-PF was moderately resistant to the depleting

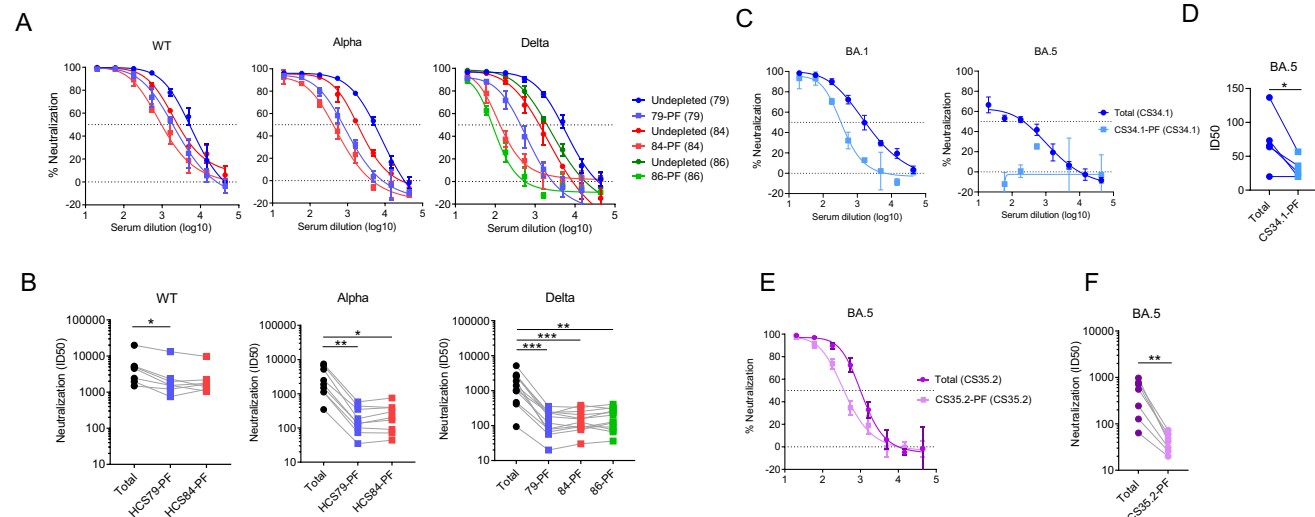

**Fig. 4 | Isolation of PF using polyclonal serum antibodies and assessment of its neutralization sensitivity. A** The neutralization profile of WT, Alpha, and Delta pseudovirus and their corresponding PFs isolated by using total IgG from COVID-19 first wave serum 79, 84, or 86. The undepleted represents the mock-depleted virus population as stated in the legend of Fig. 2. 79-PF, 84-PF, and 86-PF are the PFs obtained by depleting the pseudovirus by IgG from serum 79, 84, or 86, respectively. The neutralization of the total population of pseudovirus and its PF was assessed against the same serum that was used for virus depletion and mentioned in parentheses. Data are presented as mean values +/− SD. **B** Neutralization sensitivity of total virus and PFs in A to human convalescent heterologous sera each from different individuals from the first wave of COVID-19 (n = 7, 9, and 12 for WT, Alpha, and Delta, respectively). In this analysis, the depleting serum was not included. The statistical differences between the groups calculated by Dunn's test post

Friedman's test and adjusted *P*-values indicated as per the figure legend 3C. **C** Neutralization of undepleted BA.1 and BA.5 (blue) and their PFs (light blue) by vaccine-plus-infection serum CS34.1 from BA.1 wave. Data are presented as mean values +/− SD. **D** Neutralization sensitivity of total BA.5 and its PF to other vaccine-plus-infection sera from different individuals from BA.1 wave (n = 5). The two-tailed *p*-value (0.03) indicated as * using paired *t* test at 95% confidence level. **E** The BA.5 total population and its CS35.2-PF assessed for neutralization sensitivity to the depleting serum CS35.2, which was obtained from an individual after BA.5 infection. Data are presented as mean values +/− SD. **F** Neutralization sensitivity of undepleted BA.5 and its PF to infection-plus-vaccination sera obtained each from a distinct individual from third wave of COVID-19 (n = 7). All neutralization experiments were done in two technical repeats. The two-tailed *p*-value (0.0012) indicated as ** using the Mann-Whitney paired *t* test at 95% confidence level.

antibody, ACE2-Fc, and also to other RBD-specific nAbs, except ADG20. It appeared that the binding site of ADG20 was not sensitive to conformational alterations and recognized the epitope equally efficiently in the mixed viral population and the PFs. Except ADG20, the neutralization resistance of S309-PF and HVTR88-PF to RBD specific neutralizers was significantly increased compared to the original pseudovirus population (*P* = 0.002, *P* = 0.029 respectively) (Fig. 3C). Similarly, the S309-PF and HVTR88-PF also showed substantially reduced neutralization by ACE2-Fc, the receptor ligand that recognizes only "up" conformation of RBD, suggesting that PF may predominantly contain highly closed Spike conformation. We also examined the PF obtained by an NTD-specific antibody COVA2-17, which was partially resistant to COVA2-17 but not to the other NTD-specific nAbs COVA1-22, 4A8, and an NTD-specific nanobody S1-49 (Fig. 3A, B). The enrichment of PF by an NTD antibody had only local effects restricted to the depleting nAb epitope and did not influence the epitopes of other NTD nAbs. Notably, the depletion of WT pseudovirus by RBD nAbs did not affect the sensitivity to NTD nAbs and the vice versa, suggesting that the depletion was domain-specific (Fig. 3B). We further assessed the PF of WT for the sensitivity to human convalescent sera obtained from the first wave caused by Wuhan strain of SARS-CoV-2 as our hypothesis was that the closed spike conformation would reduce the sensitivity to serum antibodies as well. The S309-PF and HVTR88-PF showed significant resistance to sera compared to undepleted virus (*P* = 0.0008, *P* = 0.0008, respectively) (Fig. 3D). The COVA2-17-PF, however, showed no significant change in sensitivity, suggesting the lack of global antigenic effects of depletion by COVA2-17. Next, we used a peptide SIH-5, which adopts a dimeric helix-hairpin structure, competes with ACE-2 for binding, and dimerizes two spike trimers by locking RBDs in "up" conformation[32]. The total virus was highly sensitive to SIH-5, COVA2-17-PF displayed slightly reduced sensitivity, but S309-PF was completely resistant (Fig. 3E). Thus, depletion by NTD-

specific COVA2-17 did not alter virus sensitivity to SIH5, suggesting that the exposure of the epitope of COVA2-17 had no implication on RBD conformations. Overall, the data suggested that S309-PF predominantly adopted a closed spike conformation which occludes ACE-2- and SIH-5-binding sites (Fig. 3F). These observations were extended to Delta by depleting the pseudovirus population using RBD antibodies - COVA2-15, HVTR88, and S309. The PFs were characteristically resistant to depleting nAb but also to other RBD antibodies, except ADG20, displaying a broadly similar pattern as seen with WT virus (Fig. 3G, H). Similarly, Delta PF was also resistant to convalescent sera (*P* < 0.0001, *P* = 0.045, *P* = 0.028 for S309, HVTR88, and COVA2-15, respectively) (Fig. 3I).

**Polyclonal serum IgG enriches the PF that is broadly resistant to neutralization**

The select sera obtained from the first-wave convalescents were potent and showed negligible PF against WT, but more PF in Alpha, Beta, and Delta neutralization was noticed (Fig. 1E, F). We hypothesized that some virions in VOC mostly bear highly closed spikes that even polyclonal nAbs can't recognize, resulting into their escape from neutralization. Three sera with a neutralization plateau below 100% were used to extract the total IgG, the predominant class of nAbs in serum, which was then used for pseudovirus depletion. As WT had negligible PF, the loss of infectivity after depletion was greater for WT than other variants because most virions were complexed with IgG and trapped by protein-A beads (Supplementary Table S1). Nonetheless, the infectious titers of PF were in the acceptable range for the standard neutralization assay. Overall, the PFs were resistant to depleting IgG, and this trend was common in all the strains tested (Fig. 4A). The difference in neutralization sensitivity between WT and its PF was relatively smaller than other variants, probably due to the originally smaller PF present in WT, which could not be enriched efficiently. The PFs of

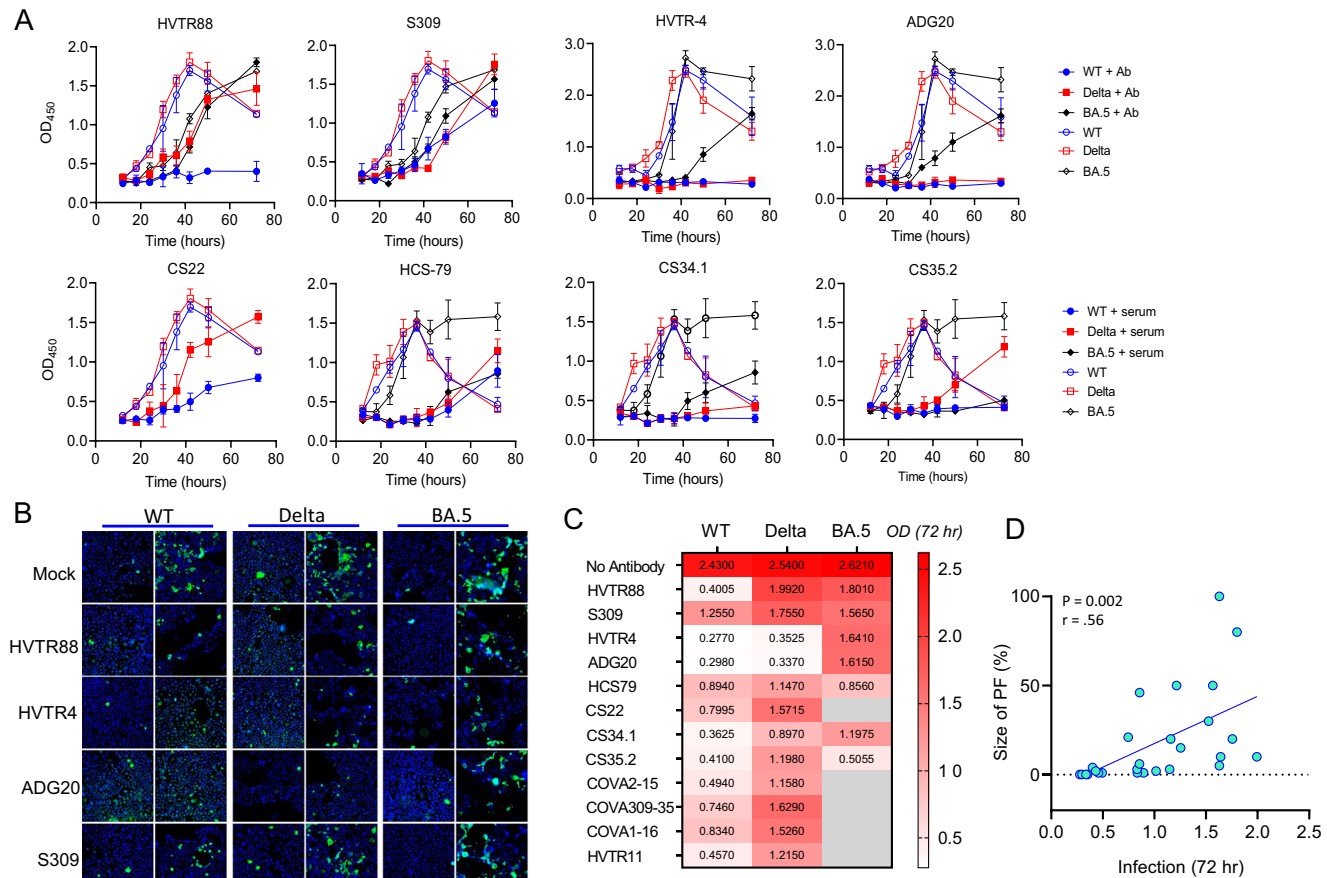

**Fig. 5 | Replication kinetics in the presence of nAb. A** Replication of authentic virus in the absence or presence of nAbs, human convalescent serum (HCS-79), vaccine serum (CS22), vaccine-plus-BA.1 infection serum (CS34.1), or vaccine-plus-BA.5 infection serum (CS35.2). The infection of Vero-TMPRSS2 cells was measured by detecting intracellular nucleocapsid protein in ELISA. All replication kinetics assays were done in two independent replicates, each with two technical repeats, showing mean values +/− SD. **B** The infection of Vero-TMPRSS2 cells shown by immunostaining of intracellular viral nucleocapsid protein after 72 h of incubation of virus with or without neutralizing antibodies, scale bar, 200 μm. The data is representative of two independent experiments. **C** The infection of SARS-CoV-2 variants is shown after 72 h of incubation as in A. The infection is depicted as OD values from ELISA. In the *No antibody* control, the growth curve was bell-shaped, therefore, the highest infection was taken from the middle of the curve. Due to near complete resistance of BA.5 to some nAbs, its replication kinetics was not tested against those mAbs, and shown in gray. **D** The correlation between the size of the PF measured from the standard neutralization assay using pseudovirus and the infection of Vero-TMPRSS2 cells by authentic virus in the presence of the nAbs. The Pearson correlation was calculated using linear regression ($n = 31$). Two-tailed $p$-value at 95% confidence level is indicated.

Alpha and Delta were more resistant to the depleting serum IgG than the corresponding undepleted virus (Fig. 4A). We next assessed the resistance of PF to other convalescent sera from the first wave. While the PF of WT was only marginally resistant, the PF of Alpha and Delta were highly resistant to sera (Fig. 4B). Thus, first wave sera showed negligible PF against WT but more against VOC.

We also included BA.1 and BA.5, representing more resistant strains emerged in the course of the pandemic. Here, we used a human serum, CS34.1, obtained from the third wave (vaccine followed by BA.1 infection) which neutralized BA.1 more effectively than BA.5. The depletion of these variants by CS34.1 IgG rendered BA.1 moderately resistant and BA.5 highly resistant to the CS34.1 IgG (Fig. 4C). The depleted BA.5 was also significantly resistant to other sera from the third wave (Fig. 4D). Subsequently, we depleted BA.5 by a serum, CS35.2, obtained from a vaccine-plus-BA.5 infected individual. The undepleted BA.5 was well sensitive to CS35.2 but its PF was resistant to it (Fig. 4E). Similarly, the CS35.2-PF of BA.5 was also highly resistant to the third-wave sera suggesting its intrinsic resistance (Fig. 4F). The data showed a trend that subsequent variants are more resistant to sera induced by earlier variants not only in terms of potency but also the efficacy of neutralization.

## PF is the source of residual infection and replication in the presence of neutralizing antibodies

Although a standard neutralization assay cannot simulate the dynamic in vivo conditions and the estimation of PF could vary between the two settings, the existence of PF itself implies its implications on replication kinetics under the antibody pressure. We hypothesized that the PF of SARS-CoV-2 is the source of replication under the nAb pressure; the complete neutralization should essentially eliminate any possibility of replication. In order to investigate PF's ability to infect cells in the presence of nAbs, three authentic viral variants - WT, Delta, and BA.5, representing major pandemic waves, and exhibiting a range of PF sizes were chosen. The viral replication in Vero-E6-TMPRSS2 cells was monitored in the presence of excess of serum or nAb concentration which achieved maximum neutralization (Fig. 5A). WT replicated in the presence of S309 and convalescent serum HCS79 as the virus had a sizable PF against them, but did not replicate in the presence of COVA2-15, HVTR4, ADG20, and serum CS35.1 because of their complete neutralization activity against WT. Similarly, Delta could replicate in the presence of COVA2-15, S309, HCS79, and slightly in the presence of CS35.1. BA.5 showed steady replication in the presence of S309, HVTR4, HCS79, CS35.1, and rapid replication occurred in the presence

of COVA2-15 and ADG20 because of the complete resistance of BA.5 to these mAbs. We also tested BA.5 in the presence of vaccine-plus-BA.1 infection serum (CS34.1) and vaccine-plus-BA.5-infection serum (CS35.2). Moderate replication of BA.5 occurred in the presence of CS34.1 but not CS35.2. The WT showed no replication with these sera, but Delta showed moderate infection with time. The infection in the presence of neutralizing agent at 72 hours was also confirmed by immunostaining assay which supported ELISA-based estimations (Fig. 5B). Overall, the replication of a SARS-CoV-2 variant in the presence of high concentrations of monoclonal or polyclonal nAbs was dependent on the size of PF against the neutralizing agent and correlated with the infection of Vero cells at 72-hours ($P = 0.004$) (Fig. 5C, D). The use of an excess concentration of nAbs ensured maximum possible neutralization of a given variant. To further confirm, we used the PF to assess its ability to replicate under neutralizing conditions. The replication of S309-PF of Delta and BA.5 was compared with their total virus populations, respectively, in the presence of S309 or a neutralizing serum HCS-79 (Supplementary Fig. S4). The basal infection by S309-PF in the presence of S309 or HCS79 was higher due to resistance to neutralization than the total population. Due to better initial infectivity, the replication was also faster compared to the total virus population (Supplementary Fig. S4A, B). Overall, the data revealed that the PF enables viral replication under immune pressure and allows its survival.

## The PF differs from total virus population in RBD glycosylation

The existence of PF in a genetically homogeneous background implied that the PTM of Spike modulates the virus neutralization. We investigated the oligomannosidic content of the virally expressed Spike using the mannose-specific lectins concanavalin A (ConA) and banana lectin (BanLec) in order to determine the major variations in glycosylation pattern. These lectins recognize high-mannose glycans and effectively neutralize SARS-CoV-2 variants[33,34]. First, we identified the key lectin-binding glycans essential for virus neutralization by mutating glycan sites (N to Q) in WT Spike. The glycans were knocked out from position 61, 234, or 709, which predominantly have mannosidic glycans, and also 331 and 343, which are differentially glycosylated, containing ~20% mannosidic glycans on the viral Spike[28,29,35]. The mutant pseudovirus lacking a glycan at 331 or 343, but not others, became completely resistant to ConA and BanLec, suggesting that these lectins mainly bind to RBD glycans for virus neutralization (Supplementary Fig. S5A, B). These results were in agreement with Nangarlia et al.[36]. Next, we confirmed the binding of BanLec to recombinant RBD protein expressed in the presence of Kifunensine (RBD-Kif), which retains mannosidic glycans by blocking glycan processing. The RBD-Kif showed strong binding to BanLec, but RBD without Kifunensine bound very poorly (Supplementary Fig. S5C). By the virus neutralization readout, these lectins could probe RBD glycans; binding to other glycan sites on Spike, if it occurs, does not cause neutralization and can't be deduced from the neutralization assay. The S309-PF and HVTR88-PF were significantly more sensitive to lectins than the total virus population ($P < 0.0001$, $P < 0.0001$, respectively), whereas no change was seen with COVA2-17-PF (Fig. 6A, B). We also depleted pseudovirus using CR3022, a control antibody, as it lacks the SARS-CoV-2 neutralizing activity and virus depletion is not expected due to lack of binding to viral Spikes in pre-fusion conformation[37]. The sensitivity of CR3022-PF to lectins was indeed similar to the undepleted virus (Fig. 6A, B). This data suggested that virus depleted by S309 and HVTR88 contained more high-mannose glycans in RBD, but other antibodies did not enrich virions in terms of glycosylation.

We next utilized a reverse strategy - depleting virions using a ConA-affinity column, in order to validate the glycan differences (Fig. 6C). The ConA-depleted WT showed significantly less inhibition by both ConA and BanLec, implying successful depletion of pseudovirus on the basis of RBD glycosylation. Whereas, Aleuria Aurantia

lectin (AAL), which binds to complex type glycans, potently neutralized the ConA-depleted virus than the total population. The lectin neutralization sensitivity confirmed that ConA-depleted virions have lower content of high-mannose glycans on RBD (Fig. 6D). The ConA-depleted WT was more sensitive to RBD-specific nAbs COVA2-15, S309, HVTR11, HVTR88, and also ACE2-Fc, but not to NTD nAbs COVA2-17 and 4A8 (Fig. 6E). The increased sensitivity to ACE2-Fc, which recognizes RBD in "up" conformation, suggested that the Spike on ConA-depleted viruses is in open conformation. Overall, the ConA-depletion supported the nAb-depletions, showing an inverse correlation between sensitivity to mannose-binding lectins and RBD nAbs.

To examine how a glycan alone affects antigenicity, RBD was recombinantly expressed in the absence or presence of Kifunensine (RBD-kif), the latter with high-mannose glycans at 331 and 343. The S309 binding to RBD-Kif was slightly reduced with a 1.7-fold increase of $K_D$ characterized by a faster dissociation rate compared to normal RBD (Fig. 6F). The unfavorable binding kinetics of S309 with recombinant RBD-Kif suggested that S309 does not prefer a mannosidic type glycan but a complex type glycan at 343 as reported earlier[1]. Thus, the virus particles having a higher mannosidic glycan at 343 may escape the neutralization by S309 and thus provide an explanation for the resistance of the S309-depleted virus. The data strongly suggested that the Spike conformation and glycosylation are interlinked properties in that the closed Spike is associated with mannosidic glycans on RBD, thereby making the virus more sensitive to lectins but less sensitive to RBD nAbs (Fig. 6G).

## The PF is distinguished by decreased spike incorporation but increased furin cleavage and compact spike conformation

The inverse correlation between sensitivity to lectin and RBD-nAb suggested the differentially glycosylated and conformational forms of Spike in the virus population. We further investigated whether this heterogeneity is stochastic or has any underlying causes. Following biosynthesis in the ER, the Spike migrates upward via the trans-Golgi network (TGN), where it is glycosylated and cleaved by furin at its characteristic polybasic furin cleavage site (FCS) between S1 and S2[38]. Since cleavage is an inherently inefficient event, our goal was to assess whether Spike's structural heterogeneity arises from differential cleavage. To analyze the furin cleavage of PFs, we used the authentic virus stock to obtain a sufficient quantity of PF for western blots, which was challenging with the transiently expressed pseudovirus. The cleavage was significantly increased in the PFs obtained via S309 and HVTR88 depletions ($P = 0.015$, $P = 0.019$, respectively) but not COVA2-17 and CR3022 depletions (Fig. 7A, B). A similar observation was made using S309-PF of the Delta authentic virus, which displayed almost complete cleavage, and the S0 protein was undetectable (Fig. 7A, C). The Delta Spike is naturally cleaved at S1/S2 to the highest degree among the SARS-CoV-2 variants, and only a minor amount of Spike remains uncleaved, which was efficiently eliminated by S309 depletion. The western blots also revealed the Spike incorporation in the virus particles as measured by the Spike/Nucleocapsid (S0/N) ratio. In contrast to the cleavage, the S0/N ratio was significantly reduced for S309-PF and HVTR88-PF ($P = 0.0079$, $P = 0.0159$, respectively), whereas no change was seen for COVA2-17-PF and CR3022-PF (Fig. 7D, E). Thus, while a fraction of virions in the total virus stock inherently resistant to S309 and HVTR88 had a greater proportion of cleavage, the Spike incorporation in these virions was modest. Enhanced cleavage was further evident from their infectivity in the presence of E64d, an inhibitor of the endosomal route of viral entry. The infection of S309-PF and HVTR88-PF was reduced by only 6%, compared to ~30% for undepleted WT. Thus, the infection of S309-PF and HVTR88-PF in the presence of E64d was substantially higher than undepleted virus (Fig. 7F). These data suggested that virions in S309-PF and HVTR88-PF were primarily activated by TMPRSS2 at the plasma membrane because of a greater number of cleaved Spikes. Furthermore, their fusion efficiency,

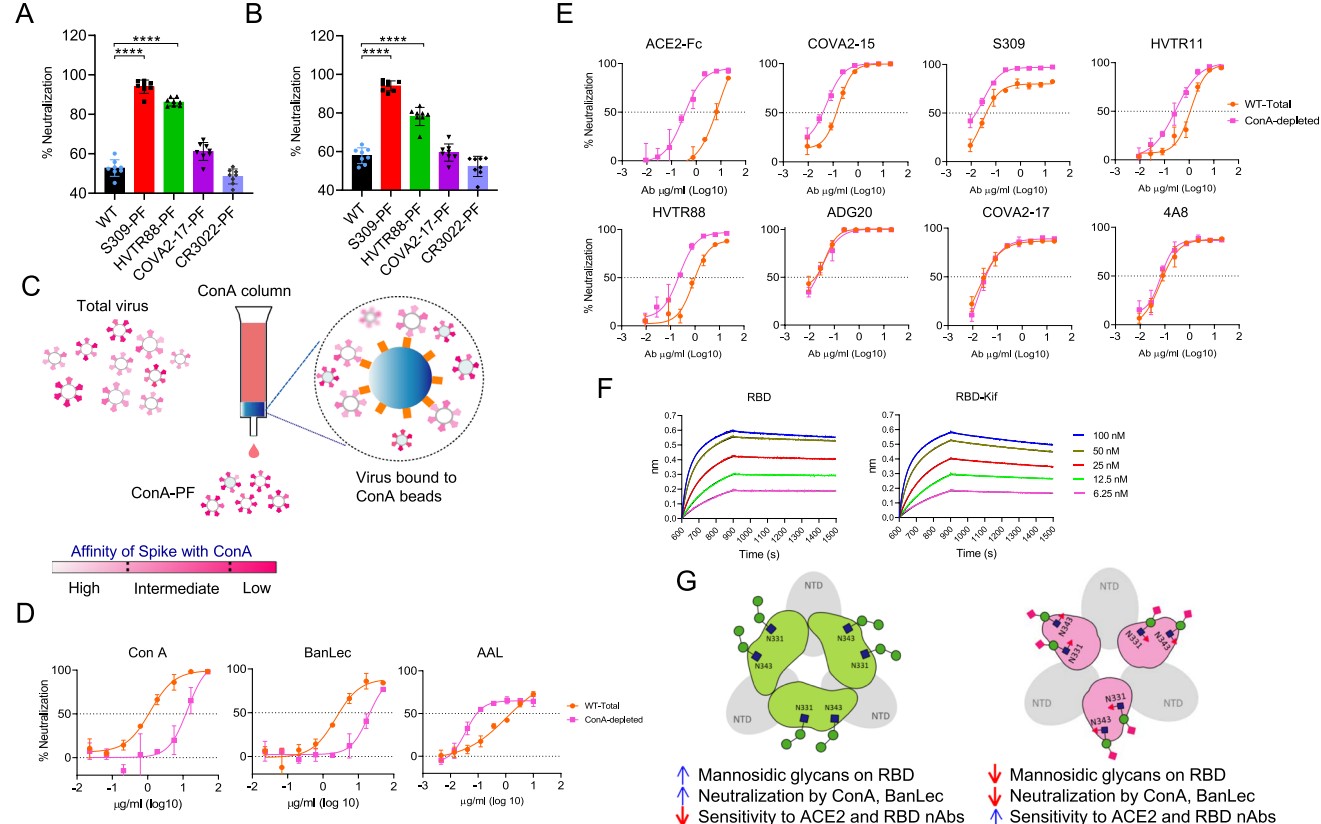

**Fig. 6 | The sensitivity of total population of WT pseudovirus and PFs to mannose-binding lectins. A, B** Lectin sensitivity of WT in the standard neutralization assay using a single concentration of lectin 5 μg/ml. Neutralization sensitivity of total pseudovirus population of WT and PFs to ConA (A) and BanLec (B). The assays were done in eight technical repeats and presented as mean values +/− SD. The statistical comparison between multiple groups was done by using Holm-Sidak's multiple comparisons test of ordinary one-way Anova. **C** The depletion of WT pseudovirus using a ConA affinity column to deplete ConA-sensitive virions. The ConA-depletion method is similar to Ab-depletion, shown in Fig. 2 and depletion is based on the principle of affinity. The ConA-PF represents the virions that passed through the ConA beads and have low affinity to ConA. **D** Neutralization of undepleted pseudovirus (Orange) and ConA-PF (Magenta) by ConA, BanLec, and complex-glycan specific Aleuria aurantia lectin (AAL). The experiment was

performed in two technical repeats and presented as mean values +/− SD.
**E** Neutralization sensitivity of undepleted WT pseudovirus (Orange) and ConA-PF (Magenta) to nAbs. All neutralization experiments were done in two technical repeats. **F** BLI sensorgrams are shown for S309 binding to recombinant RBD protein expressed in the absence or presence of kifunensine. The sensorgrams show cumulative binding of S309 at increasing concentrations to RBD after background subtraction, on the y-axis vs. time (s) on the x-axis. Each association phase was 900 s, and the dissociation was monitored for 1500 s. **G** A model of viral spike suggested by the data from pseudovirus depletions and the neutralization sensitivity to nAbs and lectins. The RBDs in "down" and "up" positions are shown in green and magenta, respectively. The RBD glycans at positions 331 and 343 are also shown; green for the mannosidic glycan on the closed spike and mixed colors for the complex type glycan on the open spike conformation.

measured by a fusion inhibitor peptide HR2-42[39], was higher than total virus population (Fig. 7H). The time required for 50% fusion, $t_{1/2}$, was 16.1 minutes for S309-PF compared to 30.3 and 25.3 minutes for COVA2-17–PF and undepleted virus, respectively (Fig. 7H). These data proved that furin cleavage is associated with the conformation and glycosylation of the Spike protein. As the presence of cleaved Spikes can facilitate rapid fusion, we assessed cell-to-cell fusion in the presence of excess nAb concentration. Indeed, the fusogenicity of variant Spikes was much higher in the presence of nAbs that exhibited a greater PF than those with minor PF against the respective spike (Supplementary Fig. S6). In another comparative analysis with WT pseudovirus and its furin site-knockout mutant (WT-GSAS), the latter was more sensitive to RBD ligands except ADG20 but less sensitive to mannose-binding lectin, supporting the conclusion that lack of furin cleavage increased RBD's propensity for "up" conformation and decreased its mannosidic glycan content (Supplementary Fig. S7). ADG20 neutralized WT and WT-GSAS equally well, suggesting that this antibody does not differentiate between "up" and "down" RBD conformations in the context of viral Spike, unlike the Spike ectodomain

used in the structural studies[2]. Overall, the data suggest the relationship between RBD glycosylation and Spike conformation mediated through furin cleavage.

## Discussion

The strength of the virus depletion technique to investigate the spike conformational heterogeneity lies in three key aspects. First, the Spike protein is in near physiological conditions and mimics a natural environment that the virus encounters in body fluids. Second, the Spike is expressed on the membrane in its native state without any artificial modifications, considering the sensitivity of conformational dynamics. Third, certain antigenic conformations can be enriched from a vast conformational spectrum to selectively assess their impact on biological properties, including but not limited to neutralization sensitivity. High-resolution cryo-EM structures of soluble spike ectodomain and dynamic conformational landscape of viral Spike have been reported[40]. While both methods inform general structure and conformations, it is not enough to explain epitope-level microheterogeneity and persistent fraction of infectivity (PF).

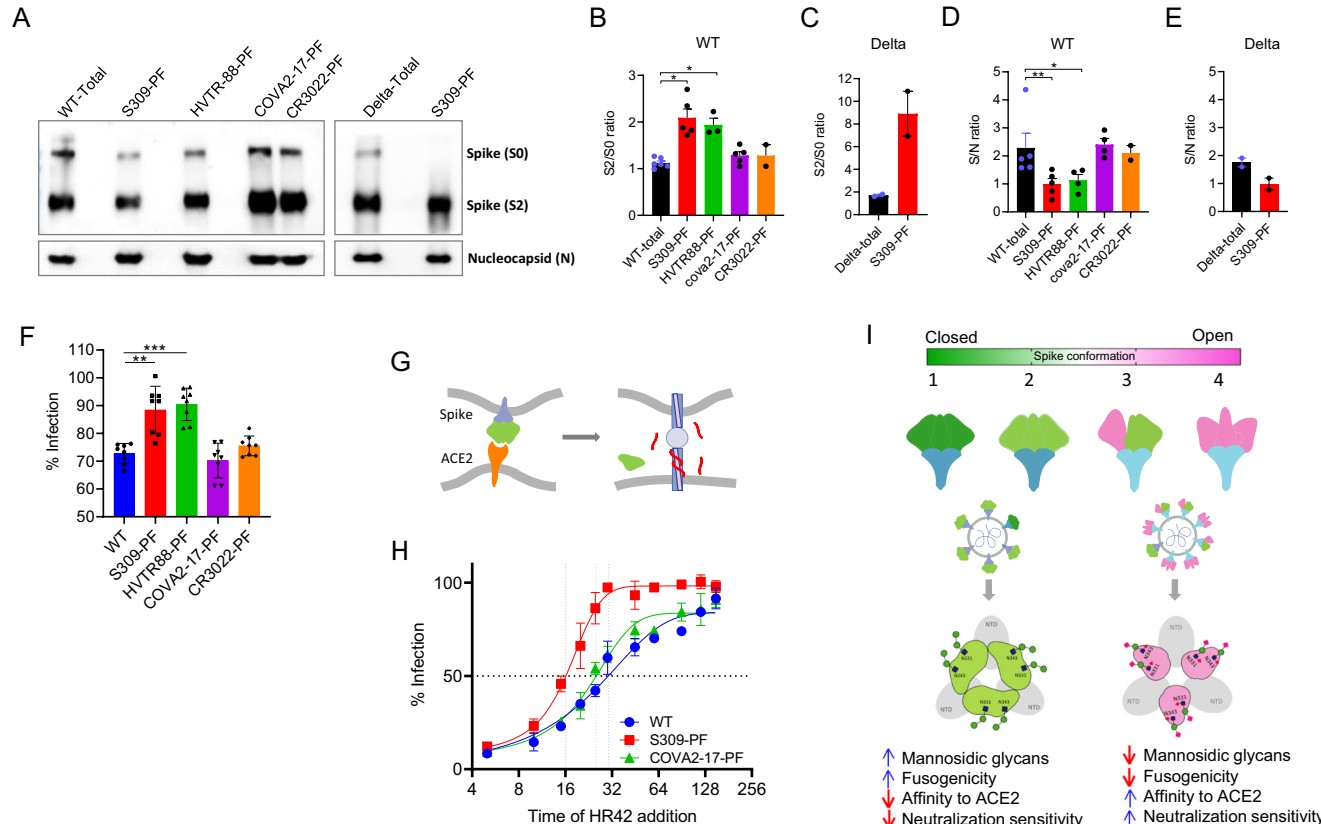

**Fig. 7 | Furin cleavage and Spike incorporation in the authentic virus particles.** Western blot analysis of authentic virus (passage-2), total and PF. **A** The Spike and nucleocapsid bands are indicated. The data was confirmed by four and two independent virus depletion experiments for WT and Delta, respectively. **B** The spike cleavage in the WT virus measured by the S2/S0 ratio. The S2/S0 ratio of undepleted virus compared with each PF using the Mann-Whitney test and two-tailed p-value denoted as * for ≤ 0.01. The data is based on the at least three biological repeats of the virus depletion experiment, creating separate blots and presented as mean values +/− SD. **C** Spike cleavage of Delta virus and its S309-PF shown as S2/S0 ratio with mean values +/− SEM. **D** Spike protein incorporation on the virions shown by the S/N ratio. The data is based on biological repeats, $n = 5$ for total virus and $n = 5, 4, 4, 2$ for S309-PF, HVTR88-PF, COVA2-17-PF, and CR3022-PF respectively. The ratio for total virus compared with S309-PF or HVTR88-PF using the Mann-Whitney test and two-tailed exact $P$-value 0.0079 indicated as ** and 0.0159 as *. **E** The S/N ratio of Delta (total) and its S309-PF derived from the western blot in A. The data is presented as mean values +/- SEM. **F.** The infection of 293T-ACE2/TMP

cells by the indicated pseudovirus in the presence of endosomal cathepsin inhibitor E64d. The experiment was done in eight technical repeats and data presented as mean values +/− SD. **G** The cartoon representation of the mechanism of inhibition of viral entry by the fusion inhibitor peptide HR42 (in red). **H** The fusion kinetics of WT pseudovirus in 293T-ACE2/TMPRSS2 cells. The entry of pre-adsorbed virions is shown as infection on the y-axis as a function of the time of addition of HR42 peptide. The $t_{1/2}$ (time for 50% fusion of adsorbed virions) depicted by a vertical dotted lines on the x-axis shown in matching colors with that of the virus. The figure represents the data of two independent experiments. **I** Differential conformational model of spike protein expressed on the virus particles based on the "up" or "down" conformation of RBD, showing four major conformations - most closed, closed, partially open, and fully open, that correspond to smFRET states described in Lu et al.[16]. The association between RBD conformation and its glycosylation is depicted by green-magenta color coding; greener corresponds to closed conformation and mannosidic glycans. The PF virions are decorated with spikes that are furin-cleaved, closed, more fusogenic, and containing mannosidic glycans on RBD.

The Spike is heavily glycosylated protein requiring to undergo folding and acquisition of glycans during the biosynthesis and migration along the secretary pathway through the ER and Golgi. The interactions with various glycan processing enzymes and chaperons create a competition for binding to Spike, resulting into the differential glycan processing and arguably the folding[41,42]. The possibility of several different glycan types at canonical sites, combined with incomplete occupancy, would result in a great deal of heterogeneity in the glycan shield, which affects neutralization. Indeed, it was recently shown by Zhang et al. that a glycan at 343 obstructs antibody-mediated neutralization and removal of this glycan by N334D substitution reduced the viral fraction that resists neutralization[30]. The glycans at positions 331 and 343 on the RBD have been assigned approximately 20% and 80% of mannosidic/hybrid and complex type glycans, respectively[28]. The S309 Fab accommodates a complex type glycan at N343, therefore, pseudovirus depletion by S309 preferentially eliminates virions rich in N343-complex glycans, leaving behind a PF

containing mannosidic glycans at this position[1]. It was evident from the greater neutralization of PF by mannose-binding lectins.

The depletion by S309 did not just reduce neutralization sensitivity to itself but also to ACE2-Fc and other RBD-specific nAbs, even though their epitope did not overlap with S309. As ACE2-Fc binds to RBM in the RBD-up conformation, poor neutralization of S309-PF by ACE2-Fc implies "down" conformation of RBD on those virions[43]. Similarly, SIH-5, which is a highly potent peptide, also binds to RBD "up" conformation and cause dimerization of inter-virion Spikes leading to virus aggregation[32], did not neutralize S309-PF virions. Thus, S309 depletion segregated the PF-containing Spike trimers in closed conformation and bearing more mannosidic glycans in RBD. When depletion was carried out in the opposite direction, i.e., by ConA, which enriches virions with more open Spike trimers containing complex type glycans in RBD, the outcome corroborated the S309-depletion. This strongly suggests a close relationship between glycosylation and spike conformation, which may have been formed during

spike processing in the ER and Golgi[44,45]. The structure-glycosylation relationship was reported for HIV glycoproteins in which the open conformation is efficiently processed for complex glycans than the closed ones[46,47]. This principle plausibly also applies to the Spike protein. The smFRET study assigned a high-FRET conformation to a highly closed conformation, which persisted even in the presence of soluble ACE2 protein[16]. This high-FRET conformation likely corresponds to Spikes present on the PF. Other RBD nAbs, HVTR88 and COVA2-15, depleted the binding site based on conformation rather than glycosylation due to their non-glycan epitope and higher affinity to open RBD conformation. The enrichment of virions was based on closed Spike, which indirectly also enriched mannosidic glycans in RBD, as evident from their higher sensitivity to mannose-binding lectins. It is possible that non-RBD glycans can also vary on the PF owing to their indirect influence on the Spike conformation, and warrants more comprehensive glycan analyzes on native viral spike in the separate study.

Is there a molecular event that controls the structural heterogeneity, or is it stochastic in nature? Here, the virus depletion offered a unique opportunity to investigate a smaller set of Ab-resistant conformations. The greater furin cleavage but reduced spike incorporation distinguished the PF from the total virus population. Therefore, we designate furin cleavage as a correlate of structural heterogeneity of the spike. The cleavage could lead to distinct folding at the local level, provide more stability to the S1-S2 interface, stabilize the closed conformation of the spike, and influence glycan processing toward high-mannose sugars, particularly in the RBD. The cleavage has been shown to be essential for infection of lung cells, transmission, and pathogenesis, therefore, the PF constitutes the frontline viruses to breakthrough host barrier for successful transmission[48–50]. The advantage with closed conformation and fewer spikes in PF would be the lack of bivalent binding of the antibody, which would drastically reduce stoichiometry needed for neutralization. Indeed, COVA2-15 showed complete neutralization and showed higher binding stoichiometry with Spike ectodomain than COVA2-17, COVA1-22, and COVA1-16, which could not achieve complete neutralization[51]. Although the cleaved and closed conformation cannot readily recognize ACE2, the infection was quick when TMPRSS2 was also present on target cells (Fig. 7H). This implies that the PF virions' receptor interaction causes quicker conformational changes that culminate in fusion, which is an advantage for transmission in the TMPRSS2-expressing upper respiratory epithelia. Poor cleavage renders the virus sensitive to neutralization, and forces it to the endosomal pathway of entry, which is much less efficient due to restriction by endosomal IFITMs[52].

In in vivo conditions, viruses may exhibit more heterogeneity of spikes owing to their production from varied cell types in the respiratory tract. The mucosal fluid has a far lower nAb concentration than serum, which would leave many virions non-neutralized when the aerosols are inhaled to the nasal linings, enhancing the probability an infection[53]. During the acute phase of infection, a higher persistent fraction (PF) may hinder rapid clearance of viremia by the immune system, thereby facilitating viral persistence and adaptation to non-respiratory tissues. This could include the exploitation of alternative entry receptors, such as neuropilin-1[19,54]. Notably, neuropilin-1 recognizes the cleaved furin site motif (RRAR), which is more commonly retained on PF virions, potentially providing an additional route for viral entry and tissue dissemination. Persistence of virus for longer periods of time may be one of the factors leading to serious immunological and pathological consequences, including long-COVID and post-acute sequelae of COVID-19 (PASC)[55]. Thus, our data provides insights into the interplay between furin cleavage and spike antigenic heterogeneity to explain the PF in neutralization, and deepens our understanding on adaptation of furin site in SARS-CoV-2 and its significance COVID-19 pathogenesis.

## Methods

The research complies with all relevant ethical regulations. All the assays involving real SARS-CoV-2 culture in a bio-safety level-3 laboratory and the pseudovirus neutralization assays were approved by the institutional biosafety committee and the institutional human ethics committee of CSIR-Institute of Microbial Technology.

### Cells

HEK293T cells were used to produce SARS-CoV-2 pseudoviruses. HEK293T expressing human ACE2 (cat # NR-52511) or ACE2 and TMPRSS2 (cat # NR-55293) were used for neutralization assays and were obtained from BEI Resources. Vero-E6-TMPRSS2 (JCRB cell bank, JCRB #1818) cells were used for the production of the authentic virus, and for replication kinetics assay, and for cell-to-cell fusion assay. All cell lines were maintained in a humidified incubator at 37 °C and 5% $CO_2$ in DMEM supplemented with 10% fetal bovine serum (FBS).

### Serum samples and ethical approval

Serum samples from the first wave were collected from individuals after 3-4 weeks after recovery from COVID-19. The vaccine sera were collected from the individuals 3-4 weeks after receiving two doses of ChAdOx1. Sera were also collected from ChAdOx1 vaccinated individuals after breakthrough infections in the third wave in February 2022. Written informed consent forms were collected from all participants before collecting blood samples for sera. The study was approved by the Institutional Ethics Committee of CSIR-Institute of Microbial Technology (Protocol number - IEC May 2020#2). All sera were limited in quantity and used in this study.

### Spike gene constructs and mutagenesis

The SARS-CoV-2 spike constructs expressing the spike protein of the Wuhan strain (WT), Alpha, Beta, Delta, BA.1, and BA.5 were codon optimized for expression in HEK293T cells and procured from GenScript Inc. All the spike constructs had D614G mutation and a 19-amino acid deletion at the C-terminal end for better cell-surface expression. The WT spike sequence with additional mutations were made by using site-directed mutagenesis by overlapping PCR fragments and ligation using the In-Fusion cloning kit (CloneTech, Inc.).

### Antibodies, soluble ACE2 protein, nanobodies, and lectins

The codon-optimized DNA sequences of heavy and light chain of S309, 4A8, and ADG20 were purchased from GenScript and cloned in pcDNA3.4. The expression constructs of COVA2-15, COVA2-17, COVA1-16, COVA309-35 were described in Brouwer et al.[56,57]. The antibodies THSC20.HVTR4, THSC20.HVTR11, THSC20.HVTR88 were described in previous studies and are referred as HVTR4, HVTR11, and HVTR88, respectively, for convenience[58,59]. ACE2-Fc-expressing plasmid was a gift from Dr. Neil King. The nanobody constructs of an NTD-specific nanobody (S1-49) and S2-specific nanobody (S2-10) were a kind gift from Dr. Michel Rout[60]. The lectin Concanavalin A (ConA) was purchased from Sigma, and banana lectin-expressing plasmid was gifted by Dr. David M. Markovitz[34]. To make the antibodies, the expi293 cells were co-transfected by heavy and light chain plasmids of antibody or ACE2-Fc in a 1:1 ratio by using polyethylenimine (PEI), and supernatants were harvested after four days post-transfection. The antibodies were purified from culture supernatants by using Protein A/G beads and finally stored in PBS for use in experiments. The nanobodies S1-49, S2-10, and banana lectin (BanLec) were expressed in E coli (Rosetta). The cells were transformed by the construct and cultured in Luria broth containing chloramphenicol and ampicillin. The cultures were induced by IPTG (Isopropyl β- d-1-thiogalactopyranoside) at the optical density of 0.4, and grown overnight at 16 °C. The cells were pelleted, reconstituted in a Tris-NaCl buffer (20 mM Tris and 250 mM NaCl), sonicated using 30% amplitude for 45 min, and centrifuged at 12,000 rpm for 45 min. The supernatant was collected and passed through the Ni/NTA

column. The protein was eluted and buffer-exchanged with PBS using dialysis tubing of 3.5 K MWCO, followed by concentration using Amicon Ultra filters.

### SARS-CoV-2 pseudovirus preparation and neutralization assay

Pseudoviruses were produced in HEK293T cells by transient transfection as described previously[61]. Briefly, cells were transfected with plasmid DNA pHIV-1 NL4·3Δenv-Luc and Spike-Δ19-D614G by using the profection mammalian transfection kit (Promega) or Polyethylenimine (PEI). To produce GFP-reporter pseudovirus, the cells were co-transfected with three plasmids - Spike-Δ19-D614G, eGFP construct, and gag-pol plasmid[61]. Virus supernatant harvested after 48 hours was passed through 0.22 μm filter, and stored at − 80 °C until further use. An aliquot from − 80 °C was thawed at room temperature and serially diluted in 96-well plate in infection medium (DMEM with 2% FBS, 100U penicillin-streptomycin) followed by the addition of 293T-hACE2-TMPRSS2 cells at a density of $1 \times 10^4$ cells/well to a final volume of 200 μL (BEI resources, NIH, Catalog No. NR-55293). The plate was incubated for 48 hours at 37 °C in a $CO_2$ incubator, and nanoluciferase activity was quantified by adding nano-luciferase substrate (Promega). The luminescence was recorded in a Cytation 5 plate reader (BioTek inc). In the case of the GFP-expressing virus, the infection was recorded as the number of GFP-expressing cells. The neutralization assays were carried out using 293T-hACE2-TMPRSS2. To assess neutralizing activity, sera were first heat- inactivated at 56 °C for 30 min and then serially diluted in growth medium starting from 1:20. The pseudovirus was incubated with serially diluted sera in a total volume of 100 μL for 1 h at 37 °C before adding 100 μL cells ($1 \times 10^4$ cells/well). The plates were further incubated for 48 h in a humidified incubator at 37 °C with 5% $CO_2$. The nanoluciferase activity was measured by using nano-Glo luciferase substrate (Promega Inc) in Cytation-5 multimode reader (BioTech Inc). The relative luminescence unit (RLU) in the absence of sera was taken as 100% infection, and percent neutralization was calculated based on the reduction of RLU. The serum dilution (ID50) or the concentration of antibody (IC50) required for 50% neutralization was calculated by the nonlinear fitting of dose-dependent inhibition curves in GraphPad Prism 8.

### Virus depletion

The pseudovirus or authentic virus depletion was done as described previously with relevant modifications[11]. Briefly, a freshly prepared authentic virus or pseudovirus stock was used for the depletion procedure. The antibody S309 (20 mg) was conjugated to 2.5 gm of activated CNBr-Sepharose 4B beads (GE Healthcare) to prepare the affinity column. 5 ml virus was incubated with S309-beads or unconjugated beads as a control with constant mixing at 37 °C for 3 h. The column was attached to the stand in an upright position until all beads were settled to form a thick layer, and clear virus supernatant was visible. The virus supernatant was passed slowly by gravity flow through the column and collected. The depleted virus ie persistent fraction (PF) was stored for further use. The beads were further washed multiple times alternately by 3 M $MgCl_2$ and 1 M glycine to remove bound virions, followed by PBS to reuse the beads for virus depletion. In the second depletion procedure, the virus stock was first incubated with antibody or serum IgG for 30 min at 37 °C. After incubation, the pseudovirus was added to the column containing an excess of protein-A/G beads (10 mg) and the virus was flowed slowly through the layer of beads to capture virus-antibody complexes and free antibodies. The flowthrough virus was again passed through another column containing an equal amount of fresh protein-A beads to remove any residual complexes or free antibodies. The final PF was frozen until further use. ConA-depletion was done by passing fresh pseudovirus through the bed of ConA beads having total 20 mg lectin. The virus supernatant was passed slowly by gravity flow with the speed of 10 ml/hour, and

flowthrough (depleted virus) collected and stored at − 80 °C. The pseudovirus depletion work was carried out in BSL-2, whereas all authentic virus depletions and neutralization assays were done in a BSL-3 laboratory.

### Pseudovirus infectivity

The infectivity of mock-depleted or antibody-depleted pseudovirus was determined in HEK293T-hACE2/TMPRSS2 cells. The p24 antigen estimation of serially diluted virus supernatants was done by using the HIV-1 p24 ELISA Kit (Abcam). After estimating the p24 of virus stocks, the virus supernatant was used for infectivity. The viruses were serially diluted in a 96-well plate in 100 μl volume, followed by the addition of 100 μl cells ($1 \times 10^4$/well). After 48 hours of incubation, the nano-luciferase activity was measured using the nano-Glo luciferase kit as a readout of infectivity (Promega).

### Virus replication kinetics

SARS-CoV-2 viruses were obtained from BEI resources and cultured in VeroE6 cells and designated as passage 1 virus (P1). Viral growth was confirmed by extensive cytopathic effect and quantitative real-time PCR. The viruses were titrated by using the same cells for plaque forming units per ml (pfu/ml) and stored at -80 °C for further use. The authentic virus replication in the presence of antibodies was assessed in VeroE6-TMPRSS2 cells. The cells were seeded in a 96-well plate ($1 \times 10^4$/well) a day before infection. The next day, the virus (0.5 MOI) was incubated with an excess concentration of antibody, which was 20-fold of the concentration required to achieve maximum neutralization in the standard pseudovirus neutralization assay. The antibody:virus mixture was then added to cells to make a total volume 200 μL, and the plate was incubated at 37 °C with 5% $CO_2$. The infection was terminated at various timepoints by discarding the culture supernatant, followed by washing cells with PBS, and then fixing by 4% PFA. After the last timepoint all wells were permeabilized by incubating with 1% saponin buffer for 1 hour. The infection was quantified by detecting N-protein by adding anti-N primary antibody, followed by anti-mouse-HRP secondary antibody. The OD value of the virus without any neutralizing agent was considered as 100% infection, and the reduction in infection was calculated with reference to this value.

### Immunofluorescence

VeroE6/TMPRSS2 cells were seeded at a density of 10,000 cells/well in 96-well optical-bottom black plates (Greiner, Cat. No. 655090). The next day, the viruses were diluted in growth media (2% FBS, penicillin-streptomycin) to 0.3−0.5 MOI and incubated for 30 minutes at 37 °C in the presence of monoclonal antibody at an excess concentration. The final concentration of antibody against a viral variant was 10-fold of the minimum concentration required to achieve maximum neutralization in the standard assay. After incubation, the virus:antibody mixture (200 μL) was added onto the cells in a 96-well plate. The infection was terminated at various time points by removing the infection media, washing cells with PBS, and fixing the cells by 4% paraformaldehyde (PFA). After fixation, the cells were permeabilized with a permeabilization buffer (PBS with 1% saponin, 1% FBS) for 30 min at room temperature. Immunofluorescence staining was performed by probing intracellular nucleocapsid using anti-nucleocapsid rabbit monoclonal (EPR24334-118). The cells were incubated with permeabilization buffer containing primary antibody (1:2000 dilution) for 1 h followed by three washes with PBS. The cells were then incubated with Alexa Fluor™ 488 secondary anti-rabbit antibody at the dilution 1:500 v/v (Invitrogen, Cat. No. A-11008) and Hoechst nucleic acid stain at the dilution of 1:2000 v/v (Thermo Fisher Scientific, Cat. No.62249) for 1 hour. The cells were washed three times with PBS, and finally, 100 μL PBS was added for preservation. The imaging was done using ImageXpress Confocal HT.ai High-Content Imaging System (Molecular Devices).

## Fusogenicity of spike protein

HEK293T cells ($4 \times 10^5$ cells/well of 6-well plate) were co-transfected by spike- and GFP-expressing constructs. In a separate 96-well plate, Vero-TMPRSS2 cells were seeded at a density of $2 \times 10^4$ cells. The next day, 293T cells were trypsinised and $5 \times 10^3$ cells were incubated with an excess concentration of antibody or serum for 30 min with intermittent shaking to ensure the binding of antibodies to the spike protein expressed on the cell membrane. The cells were then added on the monolayer of Vero-TMPRSS2 cells in a 96-well plate and plate incubated for 3 hours. The plate was imaged on a high-content imaging system for syncytia formation.

## Western blot

The fresh culture supernatants of the authentic virus were used for western blot analysis on depleted and undepleted virus. The virus supernatant was loaded onto the 20% sucrose solution in equal volumes and centrifuged at $20000 \times g$ for four hours at 4–6 °C. The supernatant was discarded, and the virus pellet was resuspended in 50 μL PBS. The virus suspension was subjected to gel electrophoresis followed by transfer of proteins to PVDF membrane (Cytiva, cat#10600029). The membrane was blocked with 5% skimmed milk in tris buffer (0.05% Tween-20) for 3 hours at room temperature. The Spike and nucleocapsid proteins were probed with rabbit anti-Spike (Ab272504) and anti-nucleocapsid (Ab271180), respectively, followed by secondary anti-rabbit IgG-HRP (BioRad, Cat#1706515). The blots were developed using chemiluminescent and chemifluorescent peroxidase substrate (Thermo Scientific, Cat#32209) and imaged on Azure 280TM (Azure Biosystems). The signal intensity of protein bands was analyzed using ImageJ software. The uncropped images with the molecular weight marker are given in the Source Data File.

## Spike protein expression and purification

The SARS-CoV-2 wild-type (Wuhan) HexaPro Spike ectodomain was produced in Expi293F™ cells. Briefly, the cells were transiently transfected at a density of $3 \times 10^6$ cells/mL in 250 mL with Spike ectodomain-expressing construct obtained from BEI resources (cat # NR-53587). The culture supernatant was harvested after four days post-transfection, centrifuged at 4000 rpm for 30 min, passed through a 0.2 μm filter and then passed through the S309 or Concanavalin A affinity columns. The captured protein was eluted by 3 M MgCl₂ buffer. The eluted proteins were extensively dialyzed in Tris-NaCl buffer (20 mM Tris-HCl, 75 mM NaCl, pH 8) and concentrated using Amicon protein concentrators (Sigma). The purity of purified protein was assessed by Blue Native-PAGE and SDS-PAGE. Spike proteins were stored at − 80 °C till further analysis. To deplete Spike ectodomain, the supernatant was first passed through the S309 column, and the flow-through supernatant containing remaining Spike protein was then passed through the ConA column. Both proteins were eluted and separately concentrated for further use. The RBD proteins were expressed in Expi293F™ cells and captured by a NiNTA column. To prevent glycan processing to complex type Kifunensine was added in the culture media after 3 hours of transfection at the final concentration of 20 μg/ml. The transfection supernatant was harvested after 4 days, and protein was purified via Ni-NTA resin (Thermo Scientific) according to the manufacturer's instructions.

## ELISA

His-tagged BanLec was coated overnight at a concentration of 10 μg/ml in 96-well Ni-NTA plate (Qiagen). The next day, the plate was washed twice with PBS and RBD protein (expressed in expi293 cells in the presence or absence of Kifunensine) was serially diluted in 1% skimmed milk and 5% FBS in PBS and added in the BanLec wells for 2 h. The plate was washed twice, and primary antibody ADG20 at a fixed concentration of 5 μg/ml was added for 1 h. The primary antibody was washed three times using PBS serially diluted in PBS containing 5% FBS and incubated with antigen for 2 h followed by 3X washing with PBS. The secondary anti-human HRP at a dilution of 1:3000 in the same buffer was then added for 45 min. The secondary antibody was washed four times with PBS containing 0.05% tween 20 and was developed by adding substrate One Step Ultra-TMB (Thermo Fisher Scientific) and incubated for 2-5 minutes until the color developed. The reaction was stopped by adding 1 N sulfuric acid, and absorbance was measured at 450 nm (Biotek).

In case of the S309 competition ELISA, serially diluted ConA starting from 10 μg/ml (75uL) was incubated with the coated Spike protein for 30 min followed by the addition of S309 or (75 μL) at the fixed concentration of 2 μg/ml making the total volume 150 μL. After 1 h incubation, the binding of S309 was detected as mentioned above.

## The biolayer interferometry (BLI)

The binding kinetics of both RBD glycoforms with S309 was studied using the Octet Red FortéBio 8-channel system (Sartorius). BLI buffer (Tris 20 mM, NaCl 75 mM) supplemented with 0.05% Tween-20 was used for all BLI experiments. Protein G biosensors were hydrated in BLI buffer for 15 min. The S309 antibody was immobilized on hydrated protein G biosensors at a concentration of 5 μg/ml and followed by an equilibration step in BLI buffer for 300 s to establish a baseline. S309-loaded sensors were immersed in wells containing different concentrations of the purified RBD proteins ranging from 100 nM to 6.25 nM (two-fold serially diluted) for 300 s (association phase) and subsequently dipped into wells containing BLI buffer for 600 s (dissociation phase). The data were reference-subtracted, and curves were fitted to a 1:1 global Langmuir model using the Octet analysis software.

## Reporting summary

Further information on research design is available in the Nature Portfolio Reporting Summary linked to this article.

## Data availability

All data supporting the findings in this study are available within the article and its Supplementary Information. The source data underlying Figs. 1–7 and Supplementary Figs. 1–7 is provided as a Source Data file. Source data are provided in this paper.

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

## Acknowledgements

We thank Dr. Paul Bieniasz for providing DNA constructs of SARS-CoV-2 Spike, HIV-1$_{NL}$ GagPol, CCNanoLuc/GFP, and pHIV-1-NL4.3-nanoLuc; Drs. Michael P. Rout for the expression construct of SARS-CoV-2 neutralizing nanobody S1-49; David M. Markovitz for banana lectin, and Jayanta Chatterjee for SIH-5 peptide. We are thankful to BEI Resources for providing the cell lines HEK293T-ACE2, HEK293T-ACE2/TMPRSS2, and viral strains SARS-CoV-2 Wuhan, Delta, and Omicron BA.5. We thank Ambey Prasad Dwivedi and Anunay Sinha for the help with western blot experiments, and Nirmal Kumar for help with high-content imaging, and T. N. C. Ramya for helpful discussion on glycan analysis. This work was funded by a grant from the Science and Engineering Research Board (IPA/2020/000168 to RPR) and the Council of Scientific and Industrial Research (CSIR) to KGT and RPR, and Bill and Melinda Gates Foundation (INV-042471) to R.V. R.V. also acknowledges infrastructural support from the following programs of the Government of India: DST-FIST, MHRD-FAST, the DBT-IISc Partnership Program, and of a JC Bose Fellowship from DST. JB acknowledges the funding support from the Bill & Melinda Gates Foundation through the GIISER South Asia grant (INV-030592). R.P. acknowledges the support of the Bill and Melinda Gates Foundation (BMGF) Grant number - INV-033578. S.K. thanks CSIR for senior research fellowship, K.K., C.S., and J.J. thank IISc, DBT, and ICMR, respectively for doctoral fellowships. RPR received the DBT-Ramalingaswami re-entry fellowship and thanks the host institute CSIR-IMTECH for providing BSL-2, -3 facilities, and instrumentation for this project.

## Author contributions

R.P.R. conceived the study. S.K., R.D., R.S., S.D., C.S., R.R., K.K., K.G.T., and R.P.R. performed research. S.K., R.D., R.S., S.D., K.K., R.P.R., and R.V. analyzed the data. R.D., C.S., and J.J. performed molecular cloning. R.D., C.S., J.J., and R.S. expressed and purified Spike and RBD proteins, antibodies, nanobodies, and banana lectin. R.P. provided COVID-19 human convalescent sera. S.D., J.B., M.J.v.G., and R.W.S. provided monoclonal nAbs isolated from COVID-19 recovered individuals. RPR wrote the first draft of the manuscript, and R.S.W., M.J.v.G., R.P., and R.V. edited the paper.

## Competing interests

The authors declare no competing interests.

## Additional information

¹CSIR-Institute of Microbial Technology (CSIR-IMTECH), Chandigarh, India. ²Molecular Biophysics Unit (MBU), Indian Institute of Science, Bengaluru, India. ³Mynvax Pvt. Ltd., 3rd Floor, Brigade MLR Centre, No.50, Vani Vilas Road, Basavanagudi, Bangalore, India. ⁴Center for Virus Research, Vaccines and Therapeutics, BRIC-Translational Health Science & Technology Institute, NCR Biotech Science Cluster, Faridabad, Haryana, India. ⁵CSIR-Institute of Genomics and Integrative Biology (CSIR-IGIB), Mall Road, Delhi, India. ⁶National Institute of Pharmaceutical Education and Research (NIPER), Mohali, Punjab, India. ⁷Department of Microbiology and Immunology, Weill Medical College of Cornell University, New York, New York, USA. ⁸Amsterdam UMC, Location University of Amsterdam, Department of Medical Microbiology and Infection Prevention, Amsterdam, The Netherlands. ⁹Amsterdam Institute for Immunology and Infectious Diseases, Amsterdam, The Netherlands. ¹⁰Jawaharlal Nehru Centre for Advanced Scientific Research, Jakkur, Bengaluru, India. ¹¹Present address: National Institute of Pharmaceutical Education and Research (NIPER), Mohali, Punjab, India. ¹²These authors contributed equally: Sahil Kumar, Rathina Delipan. ✉e-mail: rajeshringe@niper.ac.in

