## [Transparent Peer Review file · Nature Communications]

Spike conformational and glycan heterogeneity associated with furin cleavage causes incomplete neutralization of SARS-CoV-2

Corresponding Author: Dr Rajesh Ringe

Version 0:

Reviewer comments:

Reviewer #1

(Remarks to the Author)

This manuscript presents a compelling study on the role of Spike protein heterogeneity, glycosylation patterns, and furin cleavage in SARS-CoV-2's ability to evade neutralization. The findings are highly relevant to antibody-based therapies and vaccine design, offering key insights into neutralization resistance mechanisms beyond genetic mutations. While the study is intriguing, several aspects require refinement.

1. Several figure citations are incorrect or missing. Below are the necessary corrections:

- "Overall, WT was more sensitive and showed smaller PF than subsequent variants (Figure 1B, C)."

- o Correction: Should be Figure 1C, D (not 1B, C).

- "For instance, S309 neutralized WT, Alpha, Beta, and Delta with similar potency, but the efficacy of neutralization was highest for WT compared to the others (Figure 1B, C)."

- o Correction: Should be Figure 1C, D.

- "The potency of ACE2-Fc differed among the variants, but the overall efficacy of neutralization remained similar at ~100%."

- o Correction: Missing figure citation. Please add the appropriate figure.

- "Only 20/121 sera were potent, reaching the maximum neutralization plateau of WT, Alpha, Beta, and Delta (Figure 1C)."

- o Correction: Should be Figure 1E.

- "For WT, the median residual infection at lower serum dilutions plateaued below 1%, whereas for Alpha, Beta, and Delta, it did at approximately 4%, 9%, and 3%, respectively (Figure 1C, D)."

- o Correction: Should be Figure 1E, F.

2. The manuscript claims that "PF is the source of residual infection and replication in the presence of neutralizing antibodies." However, the experiments appear to have been conducted using total virus stocks rather than the isolated PF fraction. This weakens the assertion that PF alone is responsible for residual replication. To strengthen this claim, the authors should perform replication assays using isolated PF from real SARS-CoV-2 viruses, rather than relying on the total virus stock. This would provide direct evidence that PF can sustain infection and replication under neutralizing conditions.

Reviewer #2

(Remarks to the Author)

In this study, the authors describe a method of isolating the non-neutralized fraction of virions and then examine their properties. They utilize both lentiviral pseudotypes and authentic virus to characterize the "persistent fraction" (PF) representing virions non-neutralized fraction. This study benefits from examining the PF from a variety of antibodies targeting many different epitopes in spike as well as several important variants. With regards to the experiments with pseudovirus, they find that despite genetic homogeneity there is a range of post-translational modifications that influence the phenotype of the PF, especially with regards to glycosylation and furin cleavage efficiency. The findings on furin cleavage efficiency and cell-cell fusion kinetics are of particular importance and interest to the field. While this paper describes an interesting methodology for isolating the PF, the authors could do more to put in context their findings and the applications of this methodology beyond the current SARS-CoV-2 pandemic. It seems this could be a very useful technique to help with the development of broadly acting antibodies that could help to prepare us for the next pandemic.

1. This paper would benefit greatly from editing for grammar and readability.
2. The authors should describe what areas of the spike are targeted by the antibodies they are testing in the text. Some antibodies targets are described but others are not in the result sections where they are initially introduced, and this makes interpreting the results difficult. The description of the antibodies tested and why should be more organized and succinct.
3. The authors in the text frequently refer to authentic SARS-CoV-2 as “real virus”. However, in methods they refer to it as “authentic” virus. Authentic is a better term and is more standard in the field. Please edit text to utilize this term in place of “real” virus.
4. The authors spend time in both the results and the discussion discussing how the reduced neutralization by S309 and ACE2-Fc of the PF indicate a more “closed” spike. However, they also note that they do not see a decrease in neutralization of the PF with ADG20, which recognizes an epitope only present in an RBD “up” conformation (Yuan et al, PNAS, 2022). S309 binds in the RBD but can recognize the RBD in both the Up and Down configuration (Pinto et al, Nature 2020). Therefore, the argument that the neutralization escape from S309 and ACE2-Fc indicates a more “closed” spike does not fit with the neutralization data shown with ADG20, which would indicate that there are still spikes with RBD up. This is not addressed in the discussion or in any additional results sections. It would be helpful to further characterize the S309 PF given this discrepancy. Perhaps there are altered spike “up-down” kinetics that alter the binding of S309 but not ADG20. A thermostability assay with the S309 PF may help to address this. Avidity of one antibody vs. the other may also be a factor.
5. The authors need to describe the convalescent sera used more accurately in the methods. They refer to this as “first wave” sera. Please describe when exactly this sera what taken and from where geographically, as this can have significant implications for what VOC the patients in that population were exposed to.

Reviewer #3

(Remarks to the Author)

The manuscript “Conformational and glycan heterogeneity associated with furin cleavage of Spike as a cause of incomplete neutralization of SARS-CoV-2” investigates the causes for incomplete neutralization of SARS-CoV-2 by studying spike cleavage and glycosylation on virions resistant to antibody binding. The authors reveal that neutralization-resistant virus is highly cleaved between S1 and S2 subunits, is in a closed conformation and expresses more mannosidic glycans on RBD that the neutralizable virus population. I have several comments, mainly regarding employed methodology/analyses, which I am detailing below. I also would like to point out that some of the aspects investigated here have been looked at by others before, which is currently not being mentioned/discussed in this manuscript.

Main points:

1. The authors state that “studies that characterized glycans on viral Spike did not provide insights as to how glycan heterogeneity modulates viral characteristics” (lines 79-81). There is a published study that does just that: analysis of how spike glycosylation affects function and neutralization sensitivity (Zhang et al. 2024, doi: 10.1128/mbio.01672-23). In this study, a connection between spike glycosylation and incomplete neutralization with specific potent bNAbs has been reported. Removal of glycans (N343D substitution) reduced the fraction of virions that resists neutralization at high antibody concentration, specifically of a class 3 antibody. This study showed that for this antibody N343 protrudes toward the antibody binding site on the same S subunit, in line with incomplete neutralization by this antibody. This study must be discussed in the context of this work.
2. The neutralization-resistant virus fraction can only be experimentally determined when a complete neutralization curve is produced in which virus neutralization reaches a plateau. Consequentially, one must use sufficiently high antibody concentrations to reach this stable plateau of maximum neutralization, representing the non-neutralizable fraction. This is not the case for all data shown. E.g. in Figure 1 C, Ace2-Fc does not reach this plateau-phase for any of the conditions tested. It is impossible to determine the neutralization-resistant fraction from this data. Similarly, for most of the virus variants shown in Figure 1, the plateau is not observed. We do simply not know how big the neutralization resistant fraction is as one would need to test higher antibody concentrations to really determine the fraction of virus that cannot be neutralized with the tested antibodies. It is the very same for plasma neutralization in Figure 1E. Many of the variant curves do simply not reach this point, thus the PF cannot accurately be determined from these curves. While one can easily determine IC50 values from such curves, one cannot infer PF-values from any of these incomplete neutralization curves. The comparison in Figures 1D and F are not valid because of this, and this issue also affects data presented in other figures.
3. I do have doubts regarding the isolation of the “persistent fraction of infectivity (PF)”, i.e. the neutralization-resistant fraction of the virus. First, the method is solely based on binding and binding does not equal neutralization, so this really is a method not to isolate the persistent fraction of infectivity, but the non-binding fraction of virions (this should be corrected at several points throughout the manuscript. This is not a separation based on neutralization, but on binding). Importantly, the PF fractions do not always seem to have a high neutralization resistance. The authors used their presented method to isolate the presumably neutralization-resistant fraction of WT and Delta virus using 3 different antibodies. While a number of these

presumably resistant fractions indeed show increased neutralization resistance for the applied antibodies, this is not the case some conditions: E.g. Figure S3A, WT COVA2-15 and Delta HIVTR88: the neutralization-resistant fraction in the "PF" population is only marginally larger than in the total population. If a virus cannot bind an antibody, it cannot be neutralized by it. As this method is based on isolating virus populations that do not bind a given antibody, one would expect a much more pronounced phenotype, with extremely high resistance. Why is this not the case? This raises concerns about the isolation procedure. This of course has implications for all other experiments performed thereafter.

4. The relative infectivity of total population vs. Ab-sensitive-depleted population is unclear. Relative infectivity (normalized to p24) should be presented. Figure 2D: "Virus dilution" on x-axis is very misleading, as apparently the starting dilution was chosen based on p24 quantification and then serially diluted. Instead of virus dilution (which is not saying anything really if there was a pre-adjustment based on p24) the authors should plot infectivity relative to p24 (x-axis should be p24 amounts).

5. The conclusion that this study provides a "precise" explanation for the incomplete neutralization by the potent neutralizing antibodies (lines 21-23) is slightly overstated in my eyes, especially as conclusions are derived from a very limited set of antibodies (mechanistical analyses use 4 antibodies). Removing the word "precise" would be more adequate.

Minor points:

1. Please fix all wrongly called figure panels. E.g. line 119 should be Figure 1C, not B,C and line 126 should be Figure 1E not 1C, etc.. there are multiple additional errors which make it hard to find the data that is being discussed.
2. Lines 185-186: S309 recognizes a peptidoglycan epitope including a glycan at N343 and its core epitope was conserved in all the variants tested here (Figure 1B).". Figure 1B shows cell line comparisons. Entirely unclear what the authors are referring to here.
3. What is RBM? (e.g. line 188)
4. X-axes are not labeled in Figure 2C.
5. Line 118/119: "S309 neutralized WT, Alpha, Beta....with similar potency". There is almost half a log difference in IC50 for WT vs. the variants. Please rephrase.
6. Line 381: "In another comparative analysis..." Unclear what this is referring to. Published data but no reference? Or own data but not shown?

Reviewer #4

(Remarks to the Author)

This manuscript explores the phenomenon of incomplete neutralization of SARS-CoV-2, specifically the mechanisms by which certain viral particles resist neutralizing antibodies. The authors identify that these antibody-resistant virions exhibit a distinct spike (S) protein conformation. In this study, the authors developed a method to isolate the persistent fraction of infectivity (PF) and characterized the spike protein present on these virions. The neutralization resistance of PF was found to be stable and independent of the conformational equilibrium present in the pseudovirus stock. The spike proteins on PF virions were highly cleaved between the S1 and S2 subunits by the Furin protease, predominantly adopting a closed conformation, and exhibited a higher proportion of mannosylated glycans on the receptor-binding domain (RBD) compared to the total virus population. This study provides a mechanistic explanation for the incomplete neutralization observed with potent neutralizing antibodies and delineates the relationship between Furin-mediated spike cleavage, its conformational state, and glycosylation patterns.

Main Comments

1. The authors should clarify whether there is a direct relationship between high-mannose glycosylation and Furin cleavage. Which of these two factors plays a more dominant role in the formation of PF virions?
2. In Figures S2, S3, and other Figures in the main text, while PF isolated using different antibodies exhibits increased resistance to the corresponding antibodies, a relatively high level of neutralization is still observed. Notably, for COVA2-15, WT-PF remains almost fully neutralized at high antibody concentrations. How should this result be interpreted? Does it suggest a potential issue with the PF isolation method?
3. Has the presence of PF virions been observed in other cell lines, such as Vero or Caco-2 cells, which are naturally susceptible to SARS-CoV-2 infection? Does the proportion of PF virions increase in these cell types compared to 293T cells?
4. The authors should conduct in vivo infection experiments to determine whether PF virions exhibit similar infectivity in vivo as observed in vitro.

Minor Comments

1. There are numerous errors in figure citations throughout the manuscript. The authors should carefully review and correct these inaccuracies. For instance, Figure 1 appears to have been rearranged, and Figure 1F is currently not cited in the text. Additionally, there are errors in figure legends: the data source for Figure 1F should be Figure 1E, and the data source for Figure 1D should be Figure 1C.
2. The authors should include ACE2-Fc data in Figure 1D.
3. Please provide a detailed explanation for why COVA2-17-PF remains sensitive to SIH-5, whereas S309-PF is completely resistant.
4. Does Furin knockdown reduce the production of PF virions? Furthermore, how does differential glycosylation occur in viral proteins with identical genetic backgrounds? Could immunocompromised patients or individuals at high risk for severe COVID-19 be more prone to forming PF virions?
5. What is the therapeutic effect of lectins on S309-PF?

6. Since PF formation is partly driven by Furin cleavage, and previous studies have shown that Neuropilin-1 (NRP1) recognizes Furin cleavage sites and facilitates viral infection (Science. 2020 Nov 13;370(6518):856-860. doi: 10.1126/science.abd2985), what role does NRP1 play in PF-mediated infection? Moreover, if the S protein in PF state adopts a closed conformation, how does it still mediate cellular infection by recognizing the ACE2?

Version 1:

Reviewer comments:

Reviewer #1

(Remarks to the Author)

The authors have addressed all my concerns.

Reviewer #2

(Remarks to the Author)

The revisions made by the authors adequately addressed my comments and concerns in my review of the paper. These revisions have strengthened the paper and I would now recommend acceptance for publication.

Reviewer #3

(Remarks to the Author)

The authors have appropriately addressed most of my concerns. I still think that the manuscript does not appropriately describe that some aspects investigated here have been looked at by others before and that we do in fact have knowledge about the connection between glycosylation and a non-neutralizable virus fraction (Zhang et al. 2024). The presented manuscript takes a different avenue and significantly advances our previous knowledge, but the current form of the manuscript gives the impression that nothing is known about the relationship of glycans on spike and neutralization sensitivity, which is not true (lines 70-84). I strongly recommend acknowledging this prior knowledge early on in the manuscript. This will not weaken the story presented here.

Reviewer #4

(Remarks to the Author)

The authors have sufficiently addressed this reviewer's major comments with additional experiments and discussion/clarification. To this reviewer's understanding, the authors have addressed the major comments raised by other reviewers. So, this reviewer would recommend acceptance of the manuscript.

Open Access This Peer Review File is licensed under a Creative Commons Attribution 4.0 International License, which permits use, sharing, adaptation, distribution and reproduction in any medium or format, as long as you give appropriate credit to the original author(s) and the source, provide a link to the Creative Commons license, and indicate if changes were

made.

Response to reviewer comments

Reviewer #1 (Remarks to the Author):

This manuscript presents a compelling study on the role of Spike protein heterogeneity, glycosylation patterns, and furin cleavage in SARS-CoV-2's ability to evade neutralization. The findings are highly relevant to antibody-based therapies and vaccine design, offering key insights into neutralization resistance mechanisms beyond genetic mutations. While the study is intriguing, several aspects require refinement.

1. Several figure citations are incorrect or missing. Below are the necessary corrections:

- "Overall, WT was more sensitive and showed smaller PF than subsequent variants (Figure 1B, C)."

- o Correction: Should be Figure 1C, D (not 1B, C).

- "For instance, S309 neutralized WT, Alpha, Beta, and Delta with similar potency, but the efficacy of neutralization was highest for WT compared to the others (Figure 1B, C)."

- o Correction: Should be Figure 1C, D.

- "The potency of ACE2-Fc differed among the variants, but the overall efficacy of neutralization remained similar at ~100%."

- o Correction: Missing figure citation. Please add the appropriate figure.

- "Only 20/121 sera were potent, reaching the maximum neutralization plateau of WT, Alpha, Beta, and Delta (Figure 1C)."

- o Correction: Should be Figure 1E.

- "For WT, the median residual infection at lower serum dilutions plateaued below 1%, whereas for Alpha, Beta, and Delta, it did at approximately 4%, 9%, and 3%, respectively (Figure 1C, D)."

- o Correction: Should be Figure 1E, F.

Response: We regret the inconvenience caused due to this. In the revised version, we have corrected these errors.

2. The manuscript claims that "PF is the source of residual infection and replication in the presence of neutralizing antibodies." However, the experiments appear to have been conducted using total virus stocks rather than the isolated PF fraction. This weakens the assertion that PF alone is responsible for residual replication. To strengthen this claim, the authors should perform replication assays using isolated PF from real SARS-CoV-2 viruses, rather than relying on the total virus stock. This would

provide direct evidence that PF can sustain infection and replication under neutralizing conditions.

Response: The aim of the experiment was to see whether resistant virions (PF) present in the total virus population are responsible for establishing infection and slow replication in the presence of neutralizing antibodies (nAbs). In the natural condition in the presence of nAbs most virions will be neutralized but only the PF (less-sensitive to neutralization) would be able to infect the cells. The new viral progeny coming from those infected cells would again be neutralization sensitive similar to the original virus. But again, the PF in the second-generation viral progeny would infect new cells and the cycle will continue. To mimic this natural scenario, we carried out the experiment in the *in vitro* condition using authentic virus. The residual replication in the presence of various nAbs or sera therefore must be by the PF. This experiment best reflects the *in vivo* scenario and the results in Figure 5 correlate with data in Figure 2 and 4.

To address the reviewer's concern, we have carried out additional experiments using S309-PF of Delta and BA.5 authentic virus as suggested by the reviewer. The replication of equal infectious units of undepleted virus and the PF was assessed in the presence of S309 or a neutralizing serum HCS-79. As PF is less sensitive to neutralization, the first round of infection by PF in the presence of S309 or HCS-79 resulted in higher infectivity than when the total virus was used. Also, the replication of PF was faster than that of the total virus. We note that the PF in the genetically homogeneous virus stock against a neutralizing antibody is a constant as it is an attribute of Spike conformational heterogeneity. Overall, the new results verify the fact that PF is indeed a cause of the residual infection under neutralizing conditions. We have added these new data in the supplementary material (Figure S4) in the revised version and referenced it at line #319-326.

Reviewer #2 (Remarks to the Author):

In this study, the authors describe a method of isolating the non-neutralized fraction of virions and then examine their properties. They utilize both lentiviral pseudotypes and authentic virus to characterize the "persistent fraction" (PF) representing virions non-neutralized fraction. This study benefits from examining the PF from a variety of antibodies targeting many different epitopes in spike as well as several important variants. With regards to the experiments with pseudovirus, they find that despite genetic homogeneity there is a range of post-translational modifications that influence the phenotype of the PF, especially with regards to glycosylation and furin cleavage efficiency. The findings on furin cleavage efficiency and cell-cell fusion kinetics are of particular importance and interest to the field. While this paper describes an interesting methodology for isolating the PF, the authors could do more to put in context their findings and the applications of this methodology beyond the current SARS-CoV-2 pandemic. It seems this could be a very useful technique to help with the development of broadly acting antibodies that could help to prepare us for the next pandemic.

1. This paper would benefit greatly from editing for grammar and readability.

Response: We agree with the reviewer and have substantially improved the language and grammar in the revised version. We have also corrected all the figure citations.

2. The authors should describe what areas of the spike are targeted by the antibodies they are testing in the text. Some antibodies targets are described but others are not in the result sections where they are initially introduced, and this makes interpreting the results difficult. The description of the antibodies tested and why should be more organized and succinct.

Response: We have now consolidated Figure 1D in which we have mentioned the specificity of the antibody. We have also created a separate column to mention the ligand type to indicate whether it is an antibody, nanobody, or peptide. We have cited this figure panel to apprise the reader about the nature of the reagent and its specificity on the Spike protein.

The authors in the text frequently refer to authentic SARS-CoV-2 as “real virus”. However, in methods they refer to it as “authentic” virus. Authentic is a better term and is more standard in the field. Please edit text to utilize this term in place of “real” virus.

Response: We have noted the point and corrected this throughout the manuscript.

3. The authors spend time in both the results and the discussion discussing how the reduced neutralization by S309 and ACE2-Fc of the PF indicate a more “closed” spike. However, they also note that they do not see a decrease in neutralization of the PF with ADG20, which recognizes an epitope only present in an RBD “up” conformation (Yuan et al, PNAS, 2022). S309 binds in the RBD but can recognize the RBD in both the Up and Down configuration (Pinto et al, Nature 2020). Therefore, the argument that the neutralization escape from S309 and ACE2-Fc indicates a more “closed” spike does not fit with the neutralization data shown with ADG20, which would indicate that there are still spikes with RBD up. This is not addressed in the discussion or in any additional results sections. It would be helpful to further characterize the S309 PF given this discrepancy. Perhaps there are altered spike “up-down” kinetics that alter the binding of S309 but not ADG20. A thermostability assay with the S309 PF may help to address this. Avidity of one antibody vs. the other may also be a factor.

Response: We agree with the reviewer’s view regarding altered conformational dynamics of viral Spike. We add further explanation on the similar line.

In the published literature, binding of S309, hACE2, and ADG20 was examined using recombinant Spike which was an ectodomain stabilised in the pre-fusion state by 2P mutations, cleavage site knock-out mutations, and a C-terminal foldon. Although it is believed that this stabilised ectodomain largely recapitulates the Spike structure, the conformational dynamics may vary influencing the binding of the ligands.

The ligands SIH-5 peptide (Figure 3E) and ACE2-Fc have been structurally shown to bind to “up” conformation (Khatri et al Nat Chem Biol 2022, Zhang et al Mol Cell 2024). S309 binds both “up” and “down” RBD conformations on the ectodomain but in the context of viral Spike it appears to have higher affinity to “up” conformation. The neutralization pattern of S309 tracks with that of hACE2-Fc. Thus, reduced neutralization of PF by SIH-5, ACE2-Fc, and S309 strongly imply that the PF virions mostly contain “closed” Spikes, impairing their recognition by these ligands. Garrett Rappazzo et al. Science 2021 has shown that ADG-2 binds to the “up” conformation on ectodomain but also reveal that it binds with a divergent angle of approach based on structural analysis. Perhaps, due to the ability of ADG-2 to recognise epitope from diverse angle it may bind to both “up” and “down” conformations on the viral Spike and trigger conformational change to “up” conformation.

Viral thermostability is a complex issue, changes in temperature may cause other alterations in additions to shifts in the conformational equilibria between up and down RBD conformations. Hence, we did not carry out these assays. Instead, we have added a new data on the neutralization sensitivity of WT (cleaved) and WT-GSAS (uncleaved) pseudovirus. The resolution of open and closed Spike conformation is greater in this system. The WT-GSAS mutant was more sensitive to ACE2-Fc, COVA2-15, S309, HVTR88, COVA1-16, and COVA2-17 than WT (Figure S7) which bolsters the view that these ligands recognise “up” conformation more effectively. AGD20, however, neutralized WT and WT-GSAS similarly suggesting that it does not differentiate between “up” and “down” conformation.

We have added some of the aforementioned points in the Results section (line #411-419) but have refrained from over-speculating. We believe the explanation addresses reviewer’s concern.

5. The authors need to describe the convalescent sera used more accurately in the methods. They refer to this as “first wave” sera. Please describe when exactly these sera what taken and from where geographically, as this can have significant implications for what VOC the patients in that population were exposed to.

Response: We have provided more details (line 132-135) on the convalescent sera in the revised manuscript.

Reviewer #3 (Remarks to the Author):

The manuscript “Conformational and glycan heterogeneity associated with furin cleavage of Spike as a cause of incomplete neutralization of SARS-CoV-2” investigates the causes for incomplete neutralization of SARS-CoV-2 by studying spike cleavage and glycosylation on virions resistant to antibody binding. The authors reveal that neutralization-resistant virus is highly cleaved between S1 and S2 subunits, is in a closed conformation and expresses more mannosidic glycans on RBD than the neutralizable virus population. I have several comments, mainly regarding employed methodology/analyses, which I am detailing below. I also would like to point out that some

of the aspects investigated here have been looked at by others before, which is currently not being mentioned/discussed in this manuscript.

Main points:

1. The authors state that “studies that characterized glycans on viral Spike did not provide insights as to how glycan heterogeneity modulates viral characteristics” (lines 79-81). There is a published study that does just that: analysis of how spike glycosylation affects function and neutralization sensitivity (Zhang et al. 2024, doi: 10.1128/mbio.01672-23). In this study, a connection between spike glycosylation and incomplete neutralization with specific potent bNAbs has been reported. Removal of glycans (N343D substitution) reduced the fraction of virions that resists neutralization at high antibody concentration, specifically of a class 3 antibody. This study showed that for this antibody N343 protrudes toward the antibody binding site on the same S subunit, in line with incomplete neutralization by this antibody. This study must be discussed in the context of this work.

Response: We thank the reviewer for referring to a highly relevant paper which we missed in the first draft of the manuscript. The data in Zhang et al. 2024 supports our findings that the alterations in the glycosylation affect the neutralization sensitivity, and more specifically the plateau of maximum neutralization. The increase in completeness of neutralization was brought about by deliberate knock-out of the glycan. Our study, however, focussed on the natural variation of glycan types and their effect on neutralization. Thus, there are similarities but crucial difference as well. We have discussed and cited Zhang et al. 2024 paper in the revised manuscript (line # 437-440).

2. The neutralization-resistant virus fraction can only be experimentally determined when a complete neutralization curve is produced in which virus neutralization reaches a plateau. Consequentially, one must use sufficiently high antibody concentrations to reach this stable plateau of maximum neutralization, representing the non-neutralizable fraction. This is not the case for all data shown. E.g. in Figure 1 C, Ace2-Fc does not reach this plateau-phase for any of the conditions tested. It is impossible to determine the neutralization-resistant fraction from this data. Similarly, for most of the virus variants shown in Figure 1, the plateau is not observed. We do simply not know how big the neutralization resistant fraction is as one would need to test higher antibody concentrations to really determine the fraction of virus that cannot be neutralized with the tested antibodies. It is the very same for plasma neutralization in Figure 1E. Many of the variant curves do simply not reach this point, thus the PF cannot accurately be determined from these curves. While one can easily determine IC50 values from such curves, one cannot infer PF-values from any of these incomplete neutralization curves. The comparison in Figures 1D and F are not valid because of this, and this issue also affects data presented in other figures.

Response: We completely agree with the reviewer. For the uniformity purpose, 20µg/ml as maximum concentration was used for all neutralizing agents, but we admit

it's more relevant and important to achieve the clear neutralization plateau to quantify un-neutralized virus fraction. In the revised manuscript, we have replaced hACE2-Fc data in Fig 1C with a new one starting from higher concentration i.e., 60ug/ml. In addition to that, we have also replaced the neutralization curves for those ligands which could not achieve clear neutralization plateau against some variants. We increased the starting antibody concentration to 60ug/ml and assessed the variant pseudoviruses for neutralization. The data in Fig 1C is revised for the following:

- 1) COVA2-15 against Beta and Delta
- 2) COVA309-35 against Alpha
- 3) HVTR11 against WT, Alpha, Beta, Delta, and BA.5
- 4) COVA1-16 against WT, Alpha, Beta, and BA.5
- 5) S1-49 against WT, Alpha, Beta, and Delta

The new data now is suitable to estimate the PF more accurately. The data for Figure 1D has also been consolidated and is more accurate now.

While increasing the starting concentration was possible with purified reagents (monoclonal Abs, nanobody S1-49, and ACE2-Fc); unfortunately, it is not the case with convalescent sera. The lowest dilution of serum used in the neutralization assay in Figure 1E is 1:20 as per widely used methodology. Further reducing the serum dilution is technically not possible in the neutralization assay without substantially altering the media composition (first couple of wells would have high serum concentration which can alter virus infectivity). This non-uniformity affects precise estimation of dose-dependent neutralization. We confirmed this by assaying a convalescent serum with reduced dilution (1:5) and found that there is no noticeable difference between this and conventional neutralization assay. This is a general problem with less potent sera.

The goal of this experiment was to assess general difference in the maximal neutralization between WT and VOCs and the existing data did reveal the trend based on the median curve (blue) in Figure 1E. While we cannot improve the data accuracy on this technical ground, in the revised manuscript, we did reference some of the points raised by the reviewer to apprise readers about the data and its limitations (line # 141-145).

3. I do have doubts regarding the isolation of the “persistent fraction of infectivity (PF)”, i.e. the neutralization-resistant fraction of the virus. First, the method is solely based on binding and binding does not equal neutralization, so this really is a method not to isolate the persistent fraction of infectivity, but the non-binding fraction of virions (this should be corrected at several points throughout the manuscript. This is not a separation based on neutralization, but on binding). Importantly, the PF fractions do not always seem to have a high neutralization resistance. The authors used their presented method to isolate the presumably neutralization-resistant fraction of WT and Delta virus using 3 different antibodies. While a number of these presumably resistant fractions indeed show increased neutralization resistance for the applied antibodies, this is not the case some conditions: E.g. Figure S3A, WT COVA2-15 and Delta

HIVTR88: the neutralization-resistant fraction in the “PF” population is only marginally larger than in the total population. If a virus cannot bind an antibody, it cannot be neutralized by it. As this method is based on isolating virus populations that do not bind a given antibody, one would expect a much more pronounced phenotype, with extremely high resistance. Why is this not the case? This raises concerns about the isolation procedure. This of course has implications for all other experiments performed thereafter.

Response:

We respectfully disagree with the reviewer on this point. We believe the depleted virus should be designated as the “persistent fraction of infectivity” rather than the “persistent fraction of binding.”

To enrich resistant virions present in the total virus population, we employed neutralizing antibodies that exhibit varying degree of incomplete neutralization. The use of neutralizing antibodies is critical here, as it establishes a clear causal relationship — binding to the Spike protein is the cause while virus neutralization is the effect (Burton et al., *Virology* 270, 1–3, 2000). Therefore, the subset of virions that remains infectious after antibody treatment represents the population resistant to neutralization — hence, the “persistent fraction of infectivity.” If, in contrast, the depletion had been performed using non-neutralizing antibodies (targeting Spike or another viral antigen), then it might be appropriate to describe the remaining virus as a “persistent fraction of binding.” In this case, binding is uncoupled from neutralizing function, and would not correlate with loss of infectivity. Thus, because our experimental design directly links antibody binding to loss of infectivity, the terminology “persistent fraction of infectivity” more accurately captures the biological significance of the depleted and enriched fractions.

The neutralization - an outcome of antibody binding, is dependent on the affinity of Ab with viral Spike which can differ owing to the inherent heterogeneity of Spike (Figure 2). The neutralization assay registers only the net effect of a range of neutralization sensitivities of virions in the total population. Our method employed a sub-saturating concentration of antibody (Ab) to selectively enrich for virions with reduced Ab affinity. At this concentration, binding kinetics favour the interaction of Ab with spike glycoforms of higher affinity, leading to their neutralization and capture on the affinity column. Conversely, virions with low-affinity spike variants are less likely to bind and therefore pass through into the flowthrough (Fig. 2B). As a result, the flowthrough is enriched for virions with lower Ab affinity—representing a population that is moderately to highly resistant. Differential spike glycosylation likely underlies these affinity differences, as supported by our BLI analysis showing variable S309 binding to RBD glycoforms (Fig. 6F).

We acknowledge that our depletion method is not stringent enough to isolate only the most resistant virions registered as PF in a standard neutralization assay. This

limitation explains why the PF remains partially sensitive to the depleting antibody. Accordingly, we describe this approach as an “enrichment” rather than an “isolation” method for the PF. Importantly, our depletion experiments consistently demonstrated that PF enrichment is antibody-dependent. For example, the binding of virions to neutralizing antibody S309 led to more effective PF enrichment, which reflects the higher baseline PF observed against this antibody in the total virus population. In contrast, antibodies such as COVA2-15 and HVTR88 were less effective at enrichment, likely due to the much lower PFs observed for the WT and Delta variants, respectively — thereby limiting the efficiency of enrichment by these antibodies (Fig. S3). In the revised manuscript, in the Results section we have mentioned these points (line # 162-164). We also added these limitations to offer a balanced view of the method’s utility and limitations (line # 182-187)

4. The relative infectivity of total population vs. Ab-sensitive-depleted population is unclear. Relative infectivity (normalized to p24) should be presented. Figure 2D: “Virus dilution” on x-axis is very misleading, as apparently the starting dilution was chosen based on p24 quantification and then serially diluted. Instead of virus dilution (which is not saying anything really if there was a pre-adjustment based on p24) the authors should plot infectivity relative to p24 (x-axis should be p24 amounts).

Response: We have addressed this point in the revised manuscript by replacing the data with the new one. We conducted the experiment with existing stocks of the viruses starting from p24 antigen estimation followed by infectivity. The new data is similar to the old data and no significant difference in the infectivity of Total virus population versus PF is observed. We added the description of the experiment in the Methods section (line # 589-595)

5. The conclusion that this study provides a “precise” explanation for the incomplete neutralization by the potent neutralizing antibodies (lines 21-23) is slightly overstated in my eyes, especially as conclusions are derived from a very limited set of antibodies (mechanistical analyses use 4 antibodies). Removing the word “precise” would be more adequate.

Response: We have replaced the word “precise” with “possible” (line # 23).

Minor points:

1. Please fix all wrongly called figure panels. E.g. line 119 should be Figure 1C, not B,C and line 126 should be Figure 1E not 1C, etc.. there are multiple additional errors which make it hard to find the data that is being discussed.

Response: We acknowledge several errors in this figure and regret the inconvenience caused. We have now corrected the figure labels and citations.

2. Lines 185-186: S309 recognizes a peptidoglycan epitope including a glycan at N343 and its core epitope was conserved in all the variants tested here (Figure 1B).”.. Figure 1B shows cell line comparisons. Entirely unclear what the authors are referring to here.

Response: We regret this error and thanks for pointing it out. The correct citation is Figure 2C and we have corrected it in the revision (line # 207).

3. What is RBM? (e.g. line 188)

Response: We have now mentioned a long form of RBM at line # 209

4. X-axes are not labeled in Figure 2C.

Response: We have corrected this in the revision.

5. Line 118/119: "S309 neutralized WT, Alpha, Beta....with similar potency". There is almost half a log difference in IC50 for WT vs. the variants. Please rephrase.

Response: We have rephrased the description in the revised manuscript and also added a IC50 range in that context to better reflect the potency (line # 126 -128).

6. Line 381: "In another comparative analysis..." Unclear what this is referring to. Published data but no reference? Or own data but not shown?

Response: We thank reviewer for pointing this out. We had missed including this data in the figure file. We have included it in the supplementary figures (Figure S7). We revised the text regarding this at line # 411-419).

Reviewer #4 (Remarks to the Author):

This manuscript explores the phenomenon of incomplete neutralization of SARS-CoV-2, specifically the mechanisms by which certain viral particles resist neutralizing antibodies. The authors identify that these antibody-resistant virions exhibit a distinct spike (S) protein conformation. In this study, the authors developed a method to isolate the persistent fraction of infectivity (PF) and characterized the spike protein present on these virions. The neutralization resistance of PF was found to be stable and independent of the conformational equilibrium present in the pseudovirus stock. The spike proteins on PF virions were highly cleaved between the S1 and S2 subunits by the Furin protease, predominantly adopting a closed conformation, and exhibited a higher proportion of mannosylated glycans on the receptor-binding domain (RBD) compared to the total virus population. This study provides a mechanistic explanation for the incomplete neutralization observed with potent neutralizing antibodies and delineates the relationship between Furin-mediated spike cleavage, its conformational state, and glycosylation patterns.

Main Comments

1. The authors should clarify whether there is a direct relationship between high-mannose glycosylation and Furin cleavage. Which of these two factors plays a more dominant role in the formation of PF virions?

Response: We have slightly modified in the Discussion section to clarify the effect of furin cleavage on high-mannose glycosylation. As the cleavage and glycosylation are inter-linked properties, one cannot study a single factor without affecting the other. Our

data, however, do imply that there is a relationship between furin cleavage and RBD glycosylation profile. We have tried to make it clearer in the Discussion (lines 473-475) but refrained from being over-speculative.

2. In Figures S2, S3, and other Figures in the main text, while PF isolated using different antibodies exhibits increased resistance to the corresponding antibodies, a relatively high level of neutralization is still observed. Notably, for COVA2-15, WT-PF remains almost fully neutralized at high antibody concentrations. How should this result be interpreted? Does it suggest a potential issue with the PF isolation method?

Response: In our virus depletion method, we used sub-saturating Ab concentration. The binding kinetics drives the interaction of Ab molecules with the viral spike glycoforms having higher affinity to Ab. Such virions would be neutralized and more likely to be captured on the bed of affinity column. The virions with low-affinity spikes, however, would escape binding and would make their way into the flowthrough (Figure 2B). The virions in the flowthrough are more likely to contain spikes with relatively low affinity to the antibody and would be a population of moderate-to-highly resistant virions. We do acknowledge that our virus depletion procedure is not powerful enough to isolate only the most resistant virions in absolute terms which account for the persistent fraction of infectivity in the typical neutralization assay. This is exactly why the PF was still sensitive to the depleting antibody, but to a much less extent compared to the total virus population. Therefore, we describe this as a method for “enrichment” rather than “isolation” of the PF. Our depletion experiments consistently showed that PF enrichment is antibody dependent. For instance, S309 treatment resulted in greater PF enrichment due to a higher baseline PF against this antibody in the total virus population. In contrast, COVA2-15 and HVTR88 were less effective for enrichment, as the WT and Delta variants exhibited much smaller PFs against them, respectively—making enrichment technically less efficient (Figure S3).

In the revised manuscript, we described the limitations to provide a balanced perspective on this highly useful method in the field of virology (line # 182-187). The WT-PF by COVA2-15 is less enriched because there is very small PF present in the total virus population against this Ab. We mentioned it on line # 275-277).

3. Has the presence of PF virions been observed in other cell lines, such as Vero or Caco-2 cells, which are naturally susceptible to SARS-CoV-2 infection? Does the proportion of PF virions increase in these cell types compared to 293T cells?

Response: Yes, our data confirmed that the PF is evident in other cell lines which are naturally susceptible to SARS-CoV-2 infection (VeroE6 and A549). We have added that data in the revised manuscript (Figure S1D). The description of this data is given in the Results section (line # 114 -118).

4. The authors should conduct in vivo infection experiments to determine whether PF virions exhibit similar infectivity in vivo as observed in vitro.

Response: We respectfully note that the *in vivo* experiment is beyond the scope of this study as measuring the persistent fraction (PF) and assessing its significance in *in vivo* settings would constitute a separate and substantially more complex investigation. Our cell line-based data using pseudoviruses and authentic virus is compelling and clearly established the persistent fraction of infectivity. We have also thoroughly characterized the PF virions in various assays which revealed several interesting properties which were cross-validated by different approaches. While we recognize the potential importance of investigating PF under *in vivo* conditions, such studies would require significantly more sophisticated and technically demanding methodologies. At present, we do not have the resources to undertake such work. However, we fully acknowledge its relevance and intend to explore *in vivo* studies of PF as a future research direction.

Minor Comments

1. There are numerous errors in figure citations throughout the manuscript. The authors should carefully review and correct these inaccuracies. For instance, Figure 1 appears to have been rearranged, and Figure 1F is currently not cited in the text. Additionally, there are errors in figure legends: the data source for Figure 1F should be Figure 1E, and the data source for Figure 1D should be Figure 1C.

Response: We regret those errors and the inconvenience cause because of that. In the revised version, we have corrected this.

2. The authors should include ACE2-Fc data in Figure 1D.

Response: We thank the reviewer for pointing this out. We have added the ACE2-Fc data in Fig 1D. Additionally, we have revised this data in Fig 1C as the saturation phase was not clear. We have repeated the assay by starting with higher concentration of ACE2-Fc to more accurately determine the PF.

4. Please provide a detailed explanation for why COVA2-17-PF remains sensitive to SIH-5, whereas S309-PF is completely resistant.

Response: We have provided the explanation for the same on lines # 255-257.

5. Does Furin knockdown reduce the production of PF virions? Furthermore, how does differential glycosylation occur in viral proteins with identical genetic backgrounds? Could immunocompromised patients or individuals at high risk for severe COVID-19 be more prone to forming PF virions?

Response: We have added the data on furin-knockout mutant (Figure S7) and referenced it in the Results section (line # 411-419). The WT and mutant pseudoviruses were tested in neutralization assay for the sensitivity to antibodies of different specificities. The cleavage knock-out reduced the PF for S309, HVTR88, COVA1-16 and COVA2-17 compared to its cleaved counterpart. This data bolster other data that suggests that the cleavage has significant influence on neutralization and the size of the PF.

The differential glycosylation occurs during the biosynthesis. The glycan processing is modulated by the accessibility of canonical potential N-linked glycan (PNG) sites present on the protein; the closed Spike structure would undergo less processing by enzymes due to reduced access of PNGs. Such PNGs are more likely to remain as high-mannose and not get processed to complex-type glycans. We have mentioned similar points in the Introduction section in the revised manuscript (line # 57-60).

Regarding immunocompromised patients or individuals at high risk, we opt not to comment on this matter as it will be too speculative without any supporting data.

What is the therapeutic effect of lectins on S309-PF?

Response: We used lectins as a tool to assess the Spike glycosylation profile on the PF vs total virus and the data revealed that S309-PF was more sensitive to mannose-binding lectins. From the therapeutic point of view, the combination of neutralizing antibody and mannose-binding lectin is expected to be more effective than a single agent. However, our manuscript does not intend to delve into therapeutic effects of lectin and we believe it is inappropriate to comment in the manuscript on this.

6. Since PF formation is partly driven by Furin cleavage, and previous studies have shown that Neuropilin-1 (NRP1) recognizes Furin cleavage sites and facilitates viral infection (Science. 2020 Nov 13;370(6518):856-860. doi: 10.1126/science.abd2985), what role does NRP1 play in PF-mediated infection? Moreover, if the S protein in PF state adopts a closed conformation, how does it still mediate cellular infection by recognizing the ACE2?

Response: We have discussed that Neuropilin-1 may favour the infection by the PF virions as they express cleaved Spikes which expose the CendR peptide (RRAR) as a binding site for Neuropilin-1. We have discussed this in the revised manuscript (line # 494-496)

Although PF virions exhibit lower affinity for ACE2, infection rates were significantly higher in cells co-expressing ACE2 and TMPRSS2. Viral entry depends on sequential proteolytic activation of the Spike protein, where TMPRSS2 efficiently cleaves the S2' site located just upstream of the fusion peptide on the furin-primed Spike. This facilitates rapid fusion at the plasma membrane—a more productive entry route compared to the endosomal pathway. Thus, despite suboptimal ACE2 binding, the enhanced post-binding processing steps driven by TMPRSS2 contribute to the overall higher infectivity of PF virions. Also, even though some Spikes may populate a time averaged closed conformation, there will still be conformational dynamics that permit ACE2 binding. We have explained this in the Discussion section of revised manuscript (line # 482–487).

Response to Reviewers' Comments

Reviewer #1 (Remarks to the Author):

The authors have addressed all my concerns.

Response: Thank you.

Reviewer #2 (Remarks to the Author):

The revisions made by the authors adequately addressed my comments and concerns in my review of the paper. These revisions have strengthened the paper and I would now recommend acceptance for publication.

Response: Thank you.

Reviewer #3 (Remarks to the Author):

The authors have appropriately addressed most of my concerns. I still think that the manuscript does not appropriately describe that some aspects investigated here have been looked at by others before and that we do in fact have knowledge about the connection between glycosylation and a non-neutralizable virus fraction (Zhang et al. 2024). The presented manuscript takes a different avenue and significantly advances our previous knowledge, but the current form of the manuscript gives the impression that nothing is known about the relationship of glycans on spike and neutralization sensitivity, which is not true (lines 70-84). I strongly recommend acknowledging this prior knowledge early on in the manuscript. This will not weaken the story presented here.

Response: We have given a particular attention to the Introduction section and we have sincerely acknowledged and cited the most relevant work published on this theme. We had already referred the said paper in the Discussion and now we also cite the same paper in the Introduction at line # 88-90. We marked these lines in blue.

Reviewer #4 (Remarks to the Author):

The authors have sufficiently addressed this reviewer's major comments with additional experiments and discussion/clarification. To this reviewer's understanding, the authors have addressed the major comments raised by other reviewers. So, this reviewer would recommend acceptance of the manuscript.

Response: Thank you.